# Partial Disentanglement via Mechanism Sparsity

**Sébastien Lachapelle**[1]                    **Simon Lacoste-Julien**[1,2]

[1]Mila & DIRO, Université de Montréal
[2]Canada CIFAR AI Chair

## Abstract

*Disentanglement via mechanism sparsity* was introduced recently as a principled approach to extract latent factors without supervision when the causal graph relating them in time is sparse, and/or when actions are observed and affect them sparsely. However, this theory applies only to ground-truth graphs satisfying a specific criterion. In this work, we introduce a generalization of this theory which applies to any ground-truth graph and specifies *qualitatively* how disentangled the learned representation is expected to be, via a new equivalence relation over models we call *consistency*. This equivalence captures which factors are expected to remain entangled and which are not based on the specific form of the ground-truth graph. We call this weaker form of identifiability *partial disentanglement*. The graphical criterion that allows *complete* disentanglement, proposed in an earlier work, can be derived as a special case of our theory. Finally, we enforce graph sparsity with *constrained optimization* and illustrate our theory and algorithm in simulations.

## 1 INTRODUCTION

The need for *robustness*, *transferability* and *explainability* in machine learning is motivating recent efforts to develop systems that capture some form of causal understanding [Pearl, 2019, Schölkopf, 2019, Goyal and Bengio, 2021]. Driven by this goal, the emerging field of *causal representation learning* [Schölkopf et al., 2021] proposes methods that attempt to reconcile the strengths of deep representation learning, which excels on high-dimensional *low-level* observations like images, with the framework of causality, which offers a formal language to describe and reason about causal relationships between *high-level* variables, e.g. object positions.

The notion of *identifiability* plays a special role in this quest to more interpretability and robustness, since models that aim at both extracting the causal variables and learning their causal relationships can easily be overdetermined, thus loosing all hope of being interpretable. The name of the game is thus to come up with inductive biases that sufficiently restrict the model class to be identifiable, while remaining sufficiently expressive to model a complex environment.

Building from the identifiability analyses of the recent literature on nonlinear ICA [Hyvarinen and Morioka, 2016, 2017, Hyvärinen et al., 2019, Khemakhem et al., 2020a,b], the work of Lachapelle et al. [2022] proposed *mechanism sparsity regularization* as an inductive bias to identify the causal latent factors. The authors showed how learning without supervision simultaneously both the latent factors and the sparse causal graph relating them can induce disentanglement, as long as technical conditions are satisfied, including a novel criterion on the ground-truth causal graph. A key distinction between other works that also learn a dependency graph over latent variables [Yang et al., 2021, Yao et al., 2022] and "disentanglement via mechanism sparsity" is that, in the latter, *disentanglement is driven by sparsity regularization*, which allows to identify model classes which are usually not identifiable without this regularization.

**Contributions:** In this work, we extend the theory of *disentanglement via mechanism sparsity* introduced by Lachapelle et al. [2022]. Instead of requiring a graphical criterion to guarantee complete disentanglement, our theory applies to arbitrary ground-truth graphs and specifies *qualitatively* how disentangled the learned representation is expected to be, via a new equivalence relation over models we call *consistency* (Def. 7). This equivalence relation captures which variables are expected to remain entangled and which are not, hence the term *partial disentanglement*. This allows, for example, to precisely express the fact that we cannot typically identify the basis in which the position of an object is expressed, but can typically disentangle it from the other objects nonetheless. The graphical criterion of Lachapelle et al. [2022], which allows *complete* disentan-

*Accepted for the Causal Representation Learning workshop at the 38[th] Conference on Uncertainty in Artificial Intelligence* (UAI CRL 2022).

glement, can be derived as a special case of our theory. We also propose to enforce sparsity via *constrained optimization* instead of regularization, following Gallego-Posada et al. [2021]. We finally illustrate our theory in simulations.

Our contribution fits nicely into the framework of Ahuja et al. [2022a], which shows how, in general, the equivariances of the transition mechanisms characterize how identifiable the representation is. Lippe et al. [2022] and Ahuja et al. [2022b] consider settings similar to ours (by interpreting actions as interventions), but the former assumes the intervention targets are known and the notion of *sparse perturbation* of the latter is closer to Locatello et al. [2020]. Also, Lippe et al. [2022], Von Kügelgen et al. [2021] and Ahuja et al. [2022b] allow for a form of block disentanglement similar to our notion of partial disentanglement. We refer the reader to Lachapelle et al. [2022] for a more extensive review of the recent literature on disentanglement and nonlinear ICA.

## 2 BACKGROUND

### 2.1 A LATENT CAUSAL MODEL

This subsection is an almost exact transcription of the model exposition of Lachapelle et al. [2022] which introduced it.

We observe the realization of a sequence of $d_x$-dimensional random vectors $\{X^t\}_{t=1}^T$ and a sequence of $d_a$-dimensional auxiliary vectors [Hyvärinen et al., 2019] $\{A^t\}_{t=0}^{T-1}$. The coordinates of $A^t$ are either discrete or continuous and can potentially represent, for example, an action taken by an agent, or a one-hot vector indexing which intervention the corresponding observation was taken from. From now on, we will refer to $A^t$ as the action vector. We assume the observations $\{X^t\}$ are generated from a sequence of latent $d_z$-dimensional continuous random vectors $\{Z^t\}_{t=1}^T$ via the equation $X^t = \mathbf{f}(Z^t) + N^t$ where $N^t \sim \mathcal{N}(0, \sigma^2 I)$ are mutually independent across time and independent of all $Z^t$ and $A^t$. Throughout, we assume $d_z \leq d_x$ and that $\mathbf{f} : \mathcal{Z} \to \mathcal{X}$ is a diffeomorphism where $\mathcal{Z}$ is the support of $Z^t$ for all $t$, and $\mathcal{X} := \mathbf{f}(\mathcal{Z})$, i.e. the image of $\mathcal{Z}$ under $\mathbf{f}$. We suppose that each factor $Z_i^t$ represents interpretable information about the observation, e.g. for high-dimensional images, the coordinates $Z_i^t$ might be the position of an object, its color, or its orientation in space. We denote $Z^{\leq t} := [Z^1 \cdots Z^t] \in \mathbb{R}^{d_z \times t}$ and analogously for $Z^{<t}$ and other random vectors.

Following previous work on nonlinear ICA [Hyvärinen et al., 2019, Khemakhem et al., 2020a], we assume

$$p(z^t \mid z^{<t}, a^{<t}) = \prod_{i=1}^{d_z} p(z_i^t \mid z^{<t}, a^{<t}), \qquad (1)$$

where each $p(z_i^t \mid z^{<t}, a^{<t})$ is in the *exponential family* [Wainwright and Jordan, 2008], i.e. $p(z_i^t \mid z^{<t}, a^{<t}) \propto$

$$h_i(z_i^t) \exp\{\mathbf{T}_i(z_i^t)^\top \boldsymbol{\lambda}_i(G_i^z \odot z^{<t}, G_i^a \odot a^{<t})\}. \quad (2)$$

Note that this family includes many well-known distributions such as the Gaussian and beta distributions. In the Gaussian case, the *sufficient statistic* is $\mathbf{T}_i(z) := (z, z^2)$ and the *base measure* is $h_i(z) := \frac{1}{\sqrt{2\pi}}$. The function $\boldsymbol{\lambda}_i(G_i^z \odot z^{<t}, G_i^a \odot a^{<t})$ outputs the *natural parameter* vector for the conditional distribution and can be itself parametrized, for instance, by a multi-layer perceptron (MLP) or a recurrent neural network (RNN). Lachapelle et al. [2022] refers to the functions $\boldsymbol{\lambda}_i$ as the *mechanisms* or the *transition functions*. In the Gaussian case, the natural parameter is two-dimensional and is related to the usual parameters $\mu$ and $\sigma^2$ via the equation $(\lambda_1, \lambda_2) = (\frac{\mu}{\sigma^2}, -\frac{1}{\sigma^2})$. We will denote by $k$ the dimensionality of the natural parameter and that of the sufficient statistic (which are equal). The binary vectors $G_i^z \in \{0, 1\}^{d_z}$ and $G_i^a \in \{0, 1\}^{d_a}$ act as masks selecting the direct parents of $z_i^t$. The Hadamard product $\odot$ is applied element-wise and broadcasted along the time dimension. Let $G^z := [G_1^z \cdots G_{d_z}^z]^\top \in \mathbb{R}^{d_z \times d_z}$, $G^a := [G_1^a \cdots G_{d_a}^a]^\top \in \mathbb{R}^{d_z \times d_a}$, $G := [G^z \, G^a]$ which is the adjacency matrix of the causal graph. Indeed, (1) & (2) describes a *causal graphical model* over the unobserved variables $Z^{\leq T}$ conditioned on the auxiliary variables $A^{<T}$.

Let $\boldsymbol{\lambda}(z^{<t}, a^{<t}) \in \mathbb{R}^{k d_z}$ be the concatenation of all $\boldsymbol{\lambda}_i(G_i^z \odot z^{<t}, G_i^a \odot a^{<t})$ and similarly for $\mathbf{T}(z^t) \in \mathbb{R}^{k d_z}$. Note that $\boldsymbol{\lambda}(z^{<t}, a^{<t})$ depends on $G$, implicitly to simplify the notation.

The learnable parameters are $\theta := (\mathbf{f}, \boldsymbol{\lambda}, G)$, which induce a conditional probability distribution $\mathbb{P}_{X^{\leq T} \mid a; \theta}$ over $X^{\leq T}$, given $A^{<T} = a$. Let $\mathcal{A} \subset \mathbb{R}^{d_a}$ be the set of possible values $A^t$ can take. We assume $p(a^{<T})$ has probability mass over all $\mathcal{A}^T$. This could arise, for instance, when $A^t$ is sampled from a policy $\pi(a^t \mid z^t)$ distribution with probability mass everywhere in $\mathcal{A}$.

### 2.2 MODEL EQUIVALENCE AND COMPLETE DISENTANGLEMENT

Given how expressive the model of Sec. 2.1 is, there is no hope of fully identifying the model from observations. Fortunately, we will see that it is unnecessary to do so to maintain interpretability. We now recall notions of model equivalence from Khemakhem et al. [2020a] & Lachapelle et al. [2022]. In what follows, we overload the notation by defining $\mathbf{f}^{-1}(z^{<t}) := [\mathbf{f}^{-1}(z^1) \cdots \mathbf{f}^{-1}(z^{t-1})]$.

**Definition 1** (Linear equivalence). *Let $\mathcal{X} := \mathbf{f}(\mathcal{Z})$ and $\tilde{\mathcal{X}} := \tilde{\mathbf{f}}(\mathcal{Z})$, i.e., the image of the support of $Z^t$ under $\mathbf{f}$ and $\tilde{\mathbf{f}}$, respectively. We say $\theta$ is **linearly equivalent** to $\tilde{\theta}$, denoted $\theta \sim_{\text{lin}} \tilde{\theta}$, if and only if $\mathcal{X} = \tilde{\mathcal{X}}$ and there exists an invertible matrix $L \in \mathbb{R}^{k d_z \times k d_z}$ and vectors $b, c \in \mathbb{R}^{k d_z}$ such that*

*1. for all $x \in \mathcal{X}$,*

$$\mathbf{T}(\mathbf{f}^{-1}(x)) = L\mathbf{T}(\tilde{\mathbf{f}}^{-1}(x)) + b$$

2. *and, for all $t \in \{1, ..., T\}, x^{<t} \in \mathcal{X}^{t-1}, a^{<t} \in \mathcal{A}^t$,*

$$L^\top \boldsymbol{\lambda}(\mathbf{f}^{-1}(x^{<t}), a^{<t}) + c = \tilde{\boldsymbol{\lambda}}(\tilde{\mathbf{f}}^{-1}(x^{<t}), a^{<t}).$$

To interpret this definition, we consider the special case where $p(z^t \mid z^{<t}, a^{<t})$ follows a Gaussian distribution with variance fixed to one. In that case, $\mathbf{T}(z) := z$ and $\boldsymbol{\lambda}$ outputs the usual mean parameter $\mu$ (here, $k = 1$), and thus, the first condition above requires that one can go from the representation $\tilde{\mathbf{f}}^{-1}(x)$ to the other representation $\mathbf{f}^{-1}(x)$ via an invertible affine transformation. The second condition on $\boldsymbol{\lambda}$ and $\tilde{\boldsymbol{\lambda}}$ is analogous.

To make sure the latent factors of two different models can be interpreted in the same way, we need something stronger than linear equivalence, since the matrix $L$ can still "mix up" different latent factors. The following equivalence relation, adapted from Lachapelle et al. [2022], does not allow for mixing. Here we assume $k = 1$ to lighten the notation.

**Definition 2** ("Up to permutation" equivalence, $k = 1$). *We say two models $\theta := (\mathbf{f}, \boldsymbol{\lambda}, G)$ and $\tilde{\theta} := (\tilde{\mathbf{f}}, \tilde{\boldsymbol{\lambda}}, \tilde{G})$ are **equivalent up to permutation**, denoted $\theta \sim_{\text{perm}} \tilde{\theta}$, if and only if there exists a permutation matrix $P$ such that*

1. *$G^z = P^\top \tilde{G}^z P$ and $G^a = P^\top \tilde{G}^a$ , and*
2. *$\theta \sim_{\text{lin}} \tilde{\theta}$ (Def. 1) with $L = DP^\top$, where the matrix $D$ is invertible and diagonal.*

Coming back to the Gaussian case with a fixed variance, equivalence up to permutation means that there exists a permutation $\pi$ such that each coordinate $i$ of one representation is equal to the scaled and shifted coordinate $\pi(i)$ of the other. Lachapelle et al. [2022] defines *disentanglement* as follows (we specify "complete" to contrast with "partial" later on).

**Definition 3** (Complete disentanglement). *Given a ground-truth model $\theta$, we say a learned model $\hat{\theta}$ is **completely disentangled** when $\theta \sim_{\text{perm}} \hat{\theta}$.*

We will see later how complete disentanglement can be relaxed to something which falls between linear equivalence and permutation equivalence.

## 2.3 LINEAR IDENTIFIABILITY

Starting now, the reader should think of $\theta$ as the *ground-truth parameter* and $\hat{\theta}$ as a *learned parameter*. The following theorem is an adaptation and minor extension of Thm. 1 from Khemakhem et al. [2020a] by Lachapelle et al. [2022]. A proof can be found in the latter.

**Theorem 4** (Conditions for linear identifiability - Khemakhem et al. [2020a], Lachapelle et al. [2022]). *Suppose we have two models as described in Sec. 2.1 with parameters $\theta = (\mathbf{f}, \boldsymbol{\lambda}, G)$ and $\hat{\theta} = (\hat{\mathbf{f}}, \hat{\boldsymbol{\lambda}}, \hat{G})$ for a fixed sequence length $T$. Suppose the following assumptions hold:*

1. *For all $i \in \{1, ..., d_z\}$, the sufficient statistic $\mathbf{T}_i$ is minimal (Def. 9).*
2. *[**Sufficient variability**] There exist $(z_{(p)}, a_{(p)})_{p=0}^{kd_z}$ in their respective supports such that the $kd_z$-dimensional vectors $(\boldsymbol{\lambda}(z_{(p)}, a_{(p)}) - \boldsymbol{\lambda}(z_{(0)}, a_{(0)}))_{p=1}^{kd_z}$ are linearly independent.*

*Then, we have linear identifiability: $\mathbb{P}_{X^{\leq T}|a;\theta} = \mathbb{P}_{X^{\leq T}|a;\hat{\theta}}$ for all $a \in \mathcal{A}^T$ implies $\theta \sim_{\text{lin}} \hat{\theta}$.*

The most important assumption is sufficient variability, which states that the ground-truth transition function $\lambda$ should be "sufficiently complex".

# 3 PARTIAL DISENTANGLEMENT VIA MECHANISM SPARSITY

## 3.1 PARTIAL DISENTANGLEMENT AND CONSISTENT MODELS

We now give a very simple definition of partial disentanglement, as something which lives strictly between linear equivalence and equivalence up to permutation:

**Definition 5** (Partial disentanglement). *Given a ground-truth model $\theta$, we say a learned model $\hat{\theta}$ is **partially disentangled** when $\theta \sim_{\text{lin}} \hat{\theta}$ with $L$ having at least one zero component and $\theta \not\sim_{\text{perm}} \hat{\theta}$.*

This definition of partial disentanglement ranges from models that are almost completely entangled, i.e. those with a very dense $L$, to ones that are very close to being completely disentangled, i.e. those with a very sparse $L$. Where a learned model falls on this continuum will depend on the ground-truth graph $G$ underlying the data generating process. To specify precisely where the zero entries of $L$ will be, we will introduce a new equivalence relation over models we call *consistency*. In order to do so, we first need to define the property of $S$-*consistency* for matrices.

**Definition 6** ($S$-consistency). *Given a binary matrix $S \in \{0, 1\}^{m \times n}$, a matrix $C \in \mathbb{R}^{m \times m}$ is $S$-**consistent** when*

$$\forall i, j, \ [\mathbb{1} - S(\mathbb{1} - S)^\top]_{i,j}^+ = 0 \implies C_{i,j} = 0, \quad (3)$$

*where $[\cdot]^+ := \max\{0, \cdot\}$ and $\mathbb{1}$ is a matrix filled with ones (assuming implicitly its correct size).*

We will interpret this definition later on in Sec. 3.2.1. For now, it is enough to understand that an $S$-consistent matrix has zeros where the binary matrix $[\mathbb{1} - S(\mathbb{1} - S)^\top]^+$ has zeros. We can now define the novel *consistency equivalence relation* over models:

**Definition 7** (Consistency equivalence, $k = 1$). *We say two models $\theta := (\mathbf{f}, \boldsymbol{\lambda}, G)$ and $\tilde{\theta} := (\tilde{\mathbf{f}}, \tilde{\boldsymbol{\lambda}}, \tilde{G})$ are **consistent**,*

denoted $\theta \sim_{\mathrm{con}} \tilde{\theta}$, if and only if there exists a permutation matrix $P$ such that

1. $G^z = P^\top \tilde{G}^z P$ and $G^a = P^\top \tilde{G}^a$ , and

2. $\theta \sim_{\mathrm{lin}} \tilde{\theta}$ (Def. 1) with $L = CP^\top$, where the matrix $C$ is $G^z$-consistent, $(G^z)^\top$-consistent and $G^a$-consistent (Def. 6).

We demonstrate in App. A.2.4 that the consistency relation over models is indeed an equivalence relation, as claimed in the the above definition. This follows from the perhaps surprising fact that the set of invertible $S$-consistent matrices forms a *group* under matrix multiplication (see Thm. 20).

The equivalence $\sim_{\mathrm{perm}}$ is stronger than $\sim_{\mathrm{con}}$, since a diagonal matrix is always $S$-consistent, for any $S$. To see this, notice that $[\mathbb{1} - S(\mathbb{1} - S)^\top]^+_{i,i} = 1$ for all $S$ and $i$.

## 3.2 IDENTIFYING THE EQUIVALENCE CLASS OF CONSISTENT MODELS

We now present the main theorem of this work which can be seen as a generalization of Thm. 5 from Lachapelle et al. [2022]. It states that, under some conditions, a perfectly fitted and maximally sparse model $\hat{\theta}$ will be *consistent* to the ground-truth distribution $\theta$, i.e. $\theta \sim_{\mathrm{con}} \hat{\theta}$ (Def. 7). It means we know *qualitatively* how disentangled the learned representation is expected to be, based on the graph $G$. See App. A.2.5 for a proof.

**Theorem 8** (Disentanglement via mechanism sparsity)**.** *Suppose we have two models as described in Sec. 2.1 with parameters $\theta = (\mathbf{f}, \boldsymbol{\lambda}, G)$ and $\hat{\theta} = (\hat{\mathbf{f}}, \hat{\boldsymbol{\lambda}}, \hat{G})$ representing the same distribution, i.e. $\mathbb{P}_{X^{\leq T}|a;\theta} = \mathbb{P}_{X^{\leq T}|a;\hat{\theta}}$ for all $a \in \mathcal{A}^T$. Suppose the assumptions of Thm. 4 hold and that,*

1. *The sufficient statistic $\mathbf{T}$ is $d_z$-dimensional ($k = 1$) and is a diffeomorphism from $\mathcal{Z}$ to $\mathbf{T}(\mathcal{Z})$.*

2. *[Sufficient time-variability] The Jacobian of the ground-truth transition function $\boldsymbol{\lambda}$ with respect to $z$ varies "sufficiently", as formalized in App. A.2.5.*

3. *[Sufficient action-variability] The ground-truth transition function $\boldsymbol{\lambda}$ is affected "sufficiently strongly" by each individual action $a_\ell$, as formalized in App. A.2.5.*

4. *[Sparsity] $||\hat{G}||_0 \leq ||G||_0$.*

*Then, $\hat{\theta}$ is consistent with $\theta$, i.e. $\theta \sim_{\mathrm{con}} \hat{\theta}$ (Def. 7).*

The conclusion that $\theta \sim_{\mathrm{con}} \hat{\theta}$ means that the learned graph $\hat{G}$ is a permutation of the ground-truth graph $G$ and that the learned representation is either completely entangled, partially disentangled or completely disentangled, depending on the ground-truth graph $G$, as formalized by Def. 7.

The **first assumption** is satisfied for example by the Gaussian case with variance fixed to one since $\mathbf{T}(z) = z$ is a

diffeomorphism. Rigorous statements of the two **sufficient variability assumptions**, initially introduced by Lachapelle et al. [2022], are relayed to App. A.2.5. Intuitively, they both require that the ground-truth transition function $\boldsymbol{\lambda}$ is complex enough. We note that these sufficient variability assumptions play a role similar to the usual *faithfulness* assumption in causal discovery [Peters et al., 2017, Section 6.5]. See App A.2.6 for more. The **sparsity assumption** requires that the learned graph is at least as sparse as the ground-truth graph. In Sec. 3.3, we suggest achieving this by enforcing a sparsity constraint on $\hat{G}$.

**The graphical criterion of Lachapelle et al. [2022].** Thm. 8 can be seen as a generalization of Thm. 5 from Lachapelle et al. [2022]. The latter requires that the ground-truth graph $G$ satisfies this criterion:[1] $\forall 1 \leq i \leq d_z$,

$$\left( \bigcap_{j \in \mathbf{Ch}_i^z} \mathbf{Pa}_j^z \right) \cap \left( \bigcap_{j \in \mathbf{Pa}_i^z} \mathbf{Ch}_j^z \right) \cap \left( \bigcap_{\ell \in \mathbf{Pa}_i^a} \mathbf{Ch}_\ell^a \right) = \{i\},$$

where $\mathbf{Pa}_i^z$ and $\mathbf{Ch}_i^z$ are the sets of parents and children of node $z_i$ in $G^z$, respectively, while $\mathbf{Ch}_\ell^a$ is the set of children of $a_\ell$ in $G^a$. This assumption allows Lachapelle et al. [2022] to identify $\theta$ up to $\sim_{\mathrm{perm}}$ (complete disentanglement) instead of up to $\sim_{\mathrm{con}}$ (possibly partial disentanglement). It turns out that, when $G$ satisfies the above criterion, the set of models that are $\sim_{\mathrm{con}}$-equivalent to $\theta$ is equal to the set of models that are $\sim_{\mathrm{perm}}$-equivalent to $\theta$. Therefore, applying Thm. 8 to a ground-truth model that satisfies the graphical criterion will guarantee complete disentanglement (see Prop. 25).

### 3.2.1 An example & interpretation

We now attempt to build intuition about the equivalence $\sim_{\mathrm{con}}$ (Def. 7) and Thm. 8 by considering an example where the ground-truth $G$ is given by $G^z = \mathbf{0}$ (no temporal dependencies) and $G^a$ is given by the bottom left of Fig. 1b. In that case, what does it mean for a model $\hat{\theta}$ to be consistent with the ground-truth $\theta$? Following Def. 7, we first have that the learned graph $\hat{G}$ is the same as $G$, up to a permutation. Secondly, we have that their representations are linked via a linear transformation $L = CP^\top$ where $C$ is $G^z$-consistent, $(G^z)^\top$-consistent and $G^a$-consistent (Def. 6). Since $G^z = \mathbf{0}$, the first two consistency properties are vacuous, i.e. they do not impose anything on $C$. However, $G^a$-consistency forces $C$ to have the same zeros as the binary matrix $[\mathbb{1} - G^a(\mathbb{1} - G^a)^\top]^+$. This binary matrix is represented at the bottom right of Fig. 1b and captures qualitatively how disentangled the learned representation is expected to be (by Thm. 8). What does Thm. 8 mean in this context? Assuming the permutation $P$ from Def. 7 is the identity for simplicity, App. A.2.8 derives the following interpretation: *the ground-truth factor $z_i$ is not a function*

---

[1]This graphical criterion is a slight simplification of the one of Lachapelle et al. [2022]. Prop. 24 shows they are equivalent.

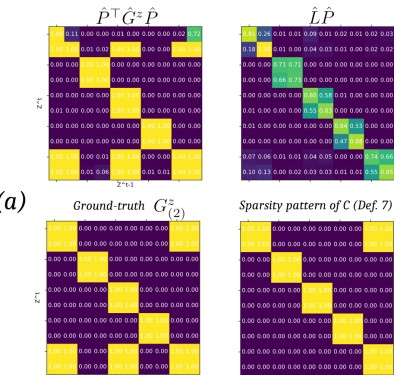 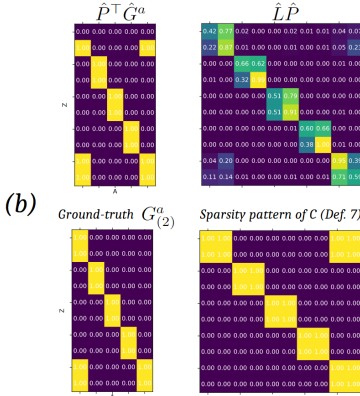

Figure 1: Typical runs on the dataset with temporal dependence **(a)** and the dataset with actions **(b)**. For both figures: **Top left:** learned graph permuted by $\hat{P}$ (the permutation found by MCC). **Bottom left:** the ground-truth graph. **Top right:** the matrix of coefficients estimated for $R$, permuted by $\hat{P}$. **Bottom right:** Expected sparsity pattern of $\hat{L}\hat{P}$, according to Thm. 8.

*of the learned factor $\hat{z}_j$ ($C_{i,j} = 0$) whenever there exists an action $a_\ell$ that targets $z_i$, but not $z_j$.*

A similar exercise can be done with different graphs $G$. For instance, consider the case where $G^a = \mathbf{0}$ (no action) and $G^z$ is given by the bottom left of Fig. 1a. In that case, $C$ will have the same zeros as the bottom right of Fig. 1a.

### 3.3 SPARSE MODEL ESTIMATION

In order to estimate from data the model presented in previous sections, we use almost the same approach as Lachapelle et al. [2022], except for how sparsity is encouraged.

To estimate the various parameters of the model, we use the well-known framework of variational autoencoders (VAEs) [Kingma and Welling, 2014] in which the decoder neural network corresponds to the mixing function $\mathbf{f}$. We consider the same approximate posterior as Lachapelle et al. [2022], that is $q(z^{\leq T} \mid x^{\leq T}, a^{<T}) := \prod_{t=1}^{T} q(z^t \mid x^t)$, where $q(z^t \mid x^t)$ is a Gaussian distribution with mean and diagonal covariance outputted by a neural network $\texttt{encoder}(x^t)$. In our experiments, the transition functions $\boldsymbol{\lambda}_i$ are parameterized by fully connected neural networks that look only at a fixed window of $s$ lagged latent variables. In all experiments, $\hat{p}(z_i^t \mid z^{<t}, a^{<t})$ is Gaussian with a learned variance that does not depend on $(z^{<t}, a^{<t})$ (see App. B.2 for details). This variational inference model induces the following evidence lower bound (ELBO) on $\log \hat{p}(x^{\leq T} \mid a^{<T})$:

$$\sum_{t=1}^{T} \mathop{\mathbb{E}}_{Z^t \sim q(\cdot \mid x^t)}[\log \hat{p}(x^t \mid Z^t)] - \mathop{\mathbb{E}}_{Z^{<t} \sim q(\cdot \mid x^{<t})} KL(q(Z^t \mid x^t) \| \hat{p}(Z^t \mid Z^{<t}, a^{<t})). \quad (4)$$

See [Lachapelle et al., 2022] for a derivation of the above.

In order to obtain $\theta \sim_{\text{con}} \hat{\theta}$. Thm. 8 suggests that, while fitting the model, we should restrict $\hat{G}$ to have at most

the same number of edges as $G$. To achieve this in practice, Lachapelle et al. [2022] introduced additional regularizing terms to the ELBO objective: $-\alpha_z \|\hat{G}^z\|_0$ and $-\alpha_a \|\hat{G}^a\|_0$. Moreover, to make the objective amenable to gradient-based optimization, they treat $\hat{G}_{i,j}^z$ and $\hat{G}_{i,\ell}^a$ as independent Bernoulli random variables with probabilities of success $\texttt{sigmoid}(\gamma_{i,j}^z)$ and $\texttt{sigmoid}(\gamma_{i,\ell}^a)$, respectively, and optimize the continuous parameters $\gamma^z$ and $\gamma^a$ using the Gumbel-Softmax gradient estimator [Jang et al., 2017, Maddison et al., 2017]. We employ a similar strategy, but instead of adding regularization terms, we add a sparsity constraint of the form $\mathbb{E}\|\hat{G}\|_0 \leq \beta$ and solve it using a variant of gradient descent-ascent on the associated Lagrangian function, as originally suggested by Gallego-Posada et al. [2021] to learn sparse neural networks. We use the python library $\texttt{Cooper}$ [Gallego-Posada and Ramirez, 2022] which implements this algorithm for PyTorch. The main advantage of the constrained approach is that the hyperparameter $\beta$, the upper bound of the constraint, is easier to interpret than the regularizer coefficients $\alpha_z$ and $\alpha_a$, which results in easier value selection, e.g. via cross-validation. Moreover, this interpretability allowed us to design a very simple schedule for the value of $\beta$: We start training with $\beta = \max_G \|G\|_0$ and linearly decrease its value until the desired number edges is reached. See App. B.2 for optimization details.

## 4 EXPERIMENTS

The goal of this section is to demonstrate empirically that Thm. 8 holds in practice, i.e. that we can identify the equivalence class of models that are consistent (Def. 7) to the ground-truth model. Our experimental setting is largely based on the one of Lachapelle et al. [2022] and our implementation is also built on their publicly available code.

**Synthetic datasets.** We used the same synthetic datasets as Lachapelle et al. [2022], but with different ground-truth graphs to highlight partially identifiable cases where com-

| Graph | Sparsity | SHD | MCC | $R_{\text{con}}$ | $R$ | Graph | Sparsity | SHD | MCC | $R_{\text{con}}$ | $R$ |
|---|---|---|---|---|---|---|---|---|---|---|---|
| $G^z_{(1)}$ | No | — | $.61_{\pm.05}$ | $.70_{\pm.07}$ | $.98_{\pm.00}$ | $G^a_{(1)}$ | No | — | $.67_{\pm.04}$ | $.80_{\pm.08}$ | $.96_{\pm.00}$ |
| | **Yes** | $\mathbf{1.2_{\pm1.8}}$ | $\mathbf{.87_{\pm.01}}$ | $\mathbf{1.0_{\pm.00}}$ | $\mathbf{1.0_{\pm.00}}$ | | **Yes** | $\mathbf{0.4_{\pm0.9}}$ | $\mathbf{.87_{\pm.03}}$ | $\mathbf{.99_{\pm.00}}$ | $\mathbf{.99_{\pm.00}}$ |
| $G^z_{(2)}$ | No | — | $.68_{\pm.03}$ | $.78_{\pm.02}$ | $.98_{\pm.00}$ | $G^a_{(2)}$ | No | — | $.69_{\pm.05}$ | $.83_{\pm.02}$ | $.95_{\pm.00}$ |
| | **Yes** | $\mathbf{5.6_{\pm5.0}}$ | $\mathbf{.86_{\pm.02}}$ | $\mathbf{.99_{\pm.01}}$ | $\mathbf{1.0_{\pm.00}}$ | | **Yes** | $\mathbf{1.6_{\pm1.7}}$ | $\mathbf{.81_{\pm.06}}$ | $\mathbf{.98_{\pm.03}}$ | $\mathbf{.99_{\pm.01}}$ |

Table 1: **Left table:** datasets with temporal dependencies. **Right table:** datasets with actions. In both tables, two different ground-truth graphs are considered (see App. B.1 for their definitions), and for each one, we compare performance with and without the sparsity constraint. For SHD, lower is better, for MCC, $R_{\text{con}}$ and $R$, higher is better. By design, we always have $0 \leq \text{MCC} \leq R_{\text{con}} \leq R \leq 1$. Metrics are averaged over 5 random initializations and "$\pm$" indicates the standard deviation.

plete disentanglement is not guaranteed by previous works. In these cases, our theory can predict qualitatively how disentangled the learned representation is expected to be, via the $\sim_{\text{con}}$-equivalence (Def. 7). We consider two types of datasets, those with temporal dependencies, and those with actions. In both types of datasets, the ground-truth decoder $\mathbf{f}$ is a neural network initialized randomly. The latent variable $Z$ and observation $X$ have dimensionality $d_z = 10$ and $d_x = 20$, respectively. For datasets with actions, $d_a = 5$. Just like in Lachapelle et al. [2022], the ground-truth $p(z^t \mid z^{<t}, a^{<t})$ is Gaussian with covariance $\sigma_z^2 I$ and a mean outputted by some function $\mu_G(z^{t-1}, a^{t-1})$. App. B.1 gives a detailed descriptions of the function $\mu_G$ for both types of datasets. We note that the model is well specified, in the sense that transition model $\hat{p}(z^t \mid z^{t-1}, a^{t-1})$ is also Gaussian with a mean outputted by a MLP. For both types of datasets, we consider two different graphs, $G^z_{(1)}$ and $G^z_{(2)}$ for the temporal type, and $G^a_{(1)}$ and $G^a_{(2)}$ for the action type. These graphs are specified in App. B.1.

**Performance metrics.** We report four metrics to verify if we can recover the correct graphical structure as well as the representation, up to the proper equivalence class.

To measure *complete* disentanglement (Def. 3), we report the *mean correlation coefficient* (MCC), which is obtained by first computing the Pearson correlation matrix $K \in \mathbb{R}^{d_z \times d_z}$ between the ground-truth representation and the learned representation ($K_{i,j}$ is the correlation between $z_i$ and $\hat{z}_j$). Then MCC $= \max_{P \in \text{permutations}} \frac{1}{d_z} \sum_{i=1}^{d_z} |(KP)_{i,i}|$. We denote by $\hat{P}$ the optimal permutation found by MCC.

To evaluate whether the learned representation is linearly equivalent to the ground-truth (Def. 1), we perform linear regression to predict the ground-truth latent factors from the learned ones, and report the mean of the Pearson correlations between the predicted ground-truth latents and the actual ones. This metric is sometimes called the *coefficient of multiple correlation*, and happens to be the square root of the better known *coefficient of determination* denoted by $R^2$. The advantage of using $R$ instead of $R^2$ is that the former is comparable to MCC, and we always have MCC $\leq R$. Let us denote by $\hat{L}$ the matrix of estimated coefficients, which

should be thought of as an estimation of $L$ in Def. 1.

To evaluate whether the learned representation is consistent to the ground-truth (Def. 7), as predicted by Thm. 8, we perform linear regression on $\hat{P}^\top \hat{z}$ while constraining the matrix of coefficient to have the same zeros as $C$ from Def. 7, and report the mean of the associated coefficients of multiple correlation, denoted by $R_{\text{con}}$. As a consequence, we have that $0 \leq \text{MCC} \leq R_{\text{con}} \leq R \leq 1$. See App. B.3 for more details on this novel metric.

**Sparsity helps.** Table 1 shows that the sparsity constraint yields significant improvement in MCC and $R_{\text{con}}$. When the sparsity constraint is used, the gap between $R_{\text{con}}$ and $R$ is very small (both are almost 1), indicating that the learned latents that were excluded from the linear regression to compute $R_{\text{con}}$ add almost no predictive power. This indicates that the learned model is consistent to the ground-truth model (Def. 7), as predicted by Thm. 8. Moreover, the gap between MCC and $R_{\text{con}}$ is always significant, indicating that the learned representations are not completely disentangled (Def. 3), but are only partially disentangled (Def. 5), as expected. The small SHD values indicates the graph is properly learned. See Fig. 1a,b to visualize typical learned graphs. In all runs using the sparsity constraint, we set the upper bound to be $\beta := ||G||_0$. In practice, $||G||_0$ is unknown and $\beta$ must be chosen, e.g. using *unsupervised disentanglement ranking* (UDR) [Duan et al., 2020].

## 5 CONCLUSION

We introduced a generalization of the theory of *disentanglement via mechanism sparsity* [Lachapelle et al., 2022] which applies to all ground-truth causal graphs $G$. We defined a novel equivalence relation over models, we named *consistency* (Def. 7), and gave conditions for when the corresponding equivalence class can be identified from observations by enforcing sparsity (Thm. 8). We showed that the equivalences "$\sim_{\text{con}}$" and "$\sim_{\text{perm}}$" coincide when $G$ satisfies the criterion of Lachapelle et al. [2022], allowing complete instead of partial disentanglement. Finally, we proposed to enforce sparsity by solving a constrained optimization problem and validated this approach on synthetic data.

## Author Contributions

**Sébastien Lachapelle** wrote the paper, performed the experiments, came up with the theoretical results and proved them. **Simon Lacoste-Julien** provided supervision that led to clarifying various aspects of this work.

## Acknowledgements

This research was partially supported by the Canada CIFAR AI Chair Program, by an IVADO excellence PhD scholarship and by a Google Focused Research award. The experiments were in part enabled by computational resources provided by Calcul Quebec and Compute Canada. Simon Lacoste-Julien is a CIFAR Associate Fellow in the Learning in Machines & Brains program.

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

# CONTENTS

# A THEORY

## A.1 MINIMAL SUFFICIENT STATISTICS FOR EXPONENTIAL FAMILIES

The following defines what a *minimal sufficient statistics* is for an exponential family. This property ensures that the parameter of an exponential family is identifiable. See for example Wainwright and Jordan [2008, p. 40] for details.

**Definition 9** (Minimal sufficient statistic). *Given a parameterized distribution in the exponential family, as in (2), we say its sufficient statistic $\mathbf{T}_i$ is minimal when there is no $v \neq 0$ such that $v^\top \mathbf{T}_i(z)$ is constant for all $z \in \mathcal{Z}$.*

## A.2 THEORY FOR DISENTANGLEMENT VIA MECHANISM SPARSITY

### A.2.1 First insight

Recall that the conditions of Thm. 4 implies that the learned model $\hat{\theta}$ is linearly equivalent to the ground-truth model $\theta$, i.e.

$$\mathbf{T}(\mathbf{f}^{-1}(x)) = L\mathbf{T}(\hat{\mathbf{f}}^{-1}(x)) + b \tag{5}$$

$$L^\top \boldsymbol{\lambda}(\mathbf{f}^{-1}(x^{<t}), a^{<t}) + c = \hat{\boldsymbol{\lambda}}(\hat{\mathbf{f}}^{-1}(x^{<t}), a^{<t}). \tag{6}$$

The following specifies an important consequence of linear identifiability. Note that this argument is taken from Lachapelle et al. [2022].

**Lemma 10.** *Assume the dimensionality of every sufficient statistics $\mathbf{T}_i$ is $k = 1$.[2] If two models $\theta := (\mathbf{f}, \boldsymbol{\lambda}, G)$ and $\hat{\theta} = (\hat{\mathbf{f}}, \hat{\boldsymbol{\lambda}}, \hat{G})$ are linearly equivalent, i.e. $\theta \sim_L \hat{\theta}$ (Def. 1), then for all $z^{<t}, a^{<t}, \tau, \vec{\epsilon}$ in their respective supports,*

$$L^\top D_z^\tau \boldsymbol{\lambda}(z^{<t}, a^{<t}) D\mathbf{T}(z^\tau)^{-1} L = D_z^\tau \hat{\boldsymbol{\lambda}}(\mathbf{v}(z^{<t}), a^{<t}) D\mathbf{T}(\mathbf{v}(z^\tau))^{-1}, \text{ and} \tag{7}$$

$$L^\top \Delta^\tau \boldsymbol{\lambda}(z^{<t}, a^{<t}, \vec{\epsilon}) = \Delta^\tau \hat{\boldsymbol{\lambda}}(\mathbf{v}(z^{<t}), a^{<t}, \vec{\epsilon}). \tag{8}$$

*where $D_z^\tau \boldsymbol{\lambda}$ and $D_z^\tau \hat{\boldsymbol{\lambda}}$ denote Jacobian matrices with respect to $z^\tau$ and $\Delta^\tau \boldsymbol{\lambda}$ and $\Delta^\tau \hat{\boldsymbol{\lambda}}$ denote matrices of partial differences with respect to $a^\tau$, i.e.*

$$\Delta^\tau \boldsymbol{\lambda}(z^{<t}, a^{<t}, \vec{\epsilon}) := [\Delta_1^\tau \boldsymbol{\lambda}(z^{<t}, a^{<t}, \epsilon_1) \dots \Delta_{d_a}^\tau \boldsymbol{\lambda}(z^{<t}, a^{<t}, \epsilon_{d_a})] \in \mathbb{R}^{d_z \times d_a}.$$

*See Equation (95) for the definition of $\Delta_\ell^\tau \boldsymbol{\lambda}(z^{<t}, a^{<t}, \epsilon_\ell)$.*

*Proof.* We can rearrange (5) to obtain

$$\hat{\mathbf{f}}^{-1}(x) = \mathbf{T}^{-1}(L^{-1}(\mathbf{T}(\mathbf{f}^{-1}(x)) - b)) \tag{9}$$

$$\hat{\mathbf{f}}^{-1} \circ \mathbf{f}(z) = \mathbf{T}^{-1}(L^{-1}(\mathbf{T}(z) - b)) \tag{10}$$

$$\mathbf{v}(z) = \mathbf{T}^{-1}(L^{-1}(\mathbf{T}(z) - b)), \tag{11}$$

where we defined $\mathbf{v} := \hat{\mathbf{f}}^{-1} \circ \mathbf{f}$. Taking the derivative of (11) w.r.t. $z$, we obtain

$$D\mathbf{v}(z) = D\mathbf{T}^{-1}(L^{-1}(\mathbf{T}(z) - b))L^{-1}D\mathbf{T}(z) \tag{12}$$

$$= D\mathbf{T}^{-1}(\mathbf{T}(\mathbf{v}(z)))L^{-1}D\mathbf{T}(z) \tag{13}$$

$$= D\mathbf{T}(\mathbf{v}(z))^{-1}L^{-1}D\mathbf{T}(z). \tag{14}$$

We can rewrite (6) as

$$L^\top \boldsymbol{\lambda}(z^{<t}, a^{<t}) + c = \hat{\boldsymbol{\lambda}}(\mathbf{v}(z^{<t}), a^{<t}). \tag{15}$$

By taking the derivative of the above equation w.r.t. $z^\tau$ for some $\tau \in \{1, ..., t-1\}$, we obtain

$$L^\top D_z^\tau \boldsymbol{\lambda}(z^{<t}, a^{<t}) = D_z^\tau \hat{\boldsymbol{\lambda}}(\mathbf{v}(z^{<t}), a^{<t}) D\mathbf{v}(z^\tau), \tag{16}$$

where we use $D_z^\tau$ to make explicit the fact that we are taking the derivative with respect to $z^\tau$. By plugging (14) in the above equation and rearranging the terms, we get the first desired equation:

$$\boxed{L^\top D_z^\tau \boldsymbol{\lambda}(z^{<t}, a^{<t}) D\mathbf{T}(z^\tau)^{-1} L = D_z^\tau \hat{\boldsymbol{\lambda}}(\mathbf{v}(z^{<t}), a^{<t}) D\mathbf{T}(\mathbf{v}(z^\tau))^{-1}.} \tag{17}$$

---

[2]This hypothesis is necessary only for (7) and not for (8).

To obtain the second equation, we take a partial difference w.r.t. $a_\ell^\tau$ (defined in (95)) on both sides of (15) to obtain

$$L^\top \Delta_\ell^\tau \boldsymbol{\lambda}(z^{<t}, a^{<t}, \epsilon) = \Delta_\ell^\tau \hat{\boldsymbol{\lambda}}(\mathbf{v}(z^{<t}), a^{<t}, \epsilon) \,, \tag{18}$$

where $\epsilon$ is some real number. We can regroup the partial differences for every $\ell \in \{1, ..., d_a\}$ and get

$$\Delta^\tau \boldsymbol{\lambda}(z^{<t}, a^{<t}, \vec{\epsilon}) := \left[ \Delta_1^\tau \boldsymbol{\lambda}(z^{<t}, a^{<t}, \epsilon_1) \ldots \Delta_{d_a}^\tau \boldsymbol{\lambda}(z^{<t}, a^{<t}, \epsilon_{d_a}) \right] \in \mathbb{R}^{d_z \times d_a} \,.$$

This allows us to rewrite (18) and obtain the second desired equation

$$\boxed{L^\top \Delta^\tau \boldsymbol{\lambda}(z^{<t}, a^{<t}, \vec{\epsilon}) = \Delta^\tau \hat{\boldsymbol{\lambda}}(\mathbf{v}(z^{<t}), a^{<t}, \vec{\epsilon}) \,.} \tag{19}$$

$\square$

Following the exposition of Lachapelle et al. [2022] to improve readability and present our results in their full generality, consider an arbitrary function of the form

$$\Lambda : \Gamma \to \mathbb{R}^{m \times n} \,, \tag{20}$$

where $\Gamma$ is some arbitrary set. Depending on the context, this function $\Lambda(\gamma)$ will correspond either to $D_z^\tau \boldsymbol{\lambda}(z^{<t}, a^{<t}) D\mathbf{T}(z^\tau)^{-1}$, where $\Gamma$ is the support of $(z^{<t}, a^{<t}, \tau)$, or $\Delta^\tau \boldsymbol{\lambda}(z^{<t}, a^{<t}, \vec{\epsilon})$, where $\Gamma$ is the support of $(z^{<t}, a^{<t}, \vec{\epsilon}, \tau)$.

By doing the following substitutions:

$$L^\top \underbrace{D_z^\tau \boldsymbol{\lambda}(z^{<t}, a^{<t}) D\mathbf{T}(z^\tau)^{-1}}_{\Lambda(\gamma)} L = \underbrace{D_z^\tau \hat{\boldsymbol{\lambda}}(\mathbf{v}(z^{<t}), a^{<t}) D\mathbf{T}(\mathbf{v}(z^\tau))^{-1}}_{\hat{\Lambda}(\gamma)} \,, \tag{21}$$

we get the equation:

$$L^\top \Lambda(\gamma) L = \hat{\Lambda}(\gamma) \,, \tag{22}$$

where the argument $\gamma \in \Gamma$ of the abstract function $\Lambda(\gamma)$ corresponds to $(z^{<t}, a^{<t}, \tau)$. We can do an analogous substitution

$$L^\top \underbrace{\Delta^\tau \boldsymbol{\lambda}(z^{<t}, a^{<t}, \vec{\epsilon})}_{\Lambda'(\gamma')} = \underbrace{\Delta^\tau \hat{\boldsymbol{\lambda}}(\mathbf{v}(z^{<t}), a^{<t}, \vec{\epsilon})}_{\hat{\Lambda}'(\gamma')} \,, \tag{23}$$

which yields

$$L^\top \Lambda'(\gamma') = \hat{\Lambda}'(\gamma') \,, \tag{24}$$

where the argument $\gamma' \in \Gamma'$ of the abstract function $\Lambda(\gamma')$ corresponds to $(z^{<t}, a^{<t}, \vec{\epsilon}, \tau)$.

**Key observation from Lachapelle et al. [2022]:** Notice how the zeros of $\Lambda(\gamma)$ and $\hat{\Lambda}(\gamma)$ corresponds to the missing edges in $G^z$ and $\hat{G}^z$, respectively, and how the zeros of $\Lambda'(\gamma')$ and $\hat{\Lambda}'(\gamma')$ corresponds to the missing edges in $G^a$ and $\hat{G}^a$, respectively. The intuition for why sparsity induce disentanglement is that enforcing sparsity of $G$ results in a sparse $\hat{\Lambda}(\gamma)$ and $\hat{\Lambda}'(\gamma')$, which will result in a sparse $L$ via equations (22) & (24). Since $L$ relates the ground-truth representation with the learned one, a sparse $L$ means a "more disentangled" representation. The lemmas and definitions of the following section make this intuition precise.

### A.2.2 Central Lemmas and Definitions

In order to formalize the intuition presented in the above section, we need to set up some notation and definitions. Many notation choices, definitions and results are taken from Lachapelle et al. [2022].

**Notation.** The $j$th column of $\Lambda(\gamma)$ and its $i$th row will be denoted as $\Lambda_{\cdot,j}(\gamma)$ and $\Lambda_{i,\cdot}(\gamma)$, respectively. For convenience, we will sometimes treat a binary vector $b$ as a set of indices $\{i \mid b_i = 1\}$ and sometimes treat a binary matrix $B$ as a set index couples $\{(i,j) \mid B_{i,j} = 1\}$. For example, this will allow us to write $B_{\cdot,j_1} \cap B_{\cdot,j_2}$, which should be understood as either the index set $\{i \mid B_{\cdot,j_1} = 1\} \cap \{i \mid B_{\cdot,j_2} = 1\}$ or the binary vector $B_{\cdot,j_1} \odot B_{\cdot,j_2}$ (where $\odot$ is the element-wise product),

depending on the context. Another example would be the complement of a binary vector $b^c$ which should be understood as either $\{i \mid b_i = 0\}$ or $\mathbb{1} - b$, where $\mathbb{1}$ denotes a vector filled with ones. The usefulness of this notation will become apparent later on.

We introduce further notations in the following definitions.

**Definition 11** (Aligned subspaces of $\mathbb{R}^m$). *Given a binary vector $b \in \{0,1\}^m$, we define*

$$\mathbb{R}^m_b := \{x \in \mathbb{R}^m \mid b_i = 0 \implies x_i = 0\}. \tag{25}$$

**Definition 12** (Aligned subspaces of $\mathbb{R}^{m \times n}$). *Given a binary matrix $B \in \{0,1\}^{m \times n}$, we define*

$$\mathbb{R}^{m \times n}_B := \{M \in \mathbb{R}^{m \times n} \mid B_{i,j} = 0 \implies M_{i,j} = 0\}. \tag{26}$$

Next, we define the *sparsity pattern* of $\Lambda$, which compactly captures which of its entries are always zero.

**Definition 13** (Sparsity pattern of $\Lambda$ [Lachapelle et al., 2022]). *The sparsity pattern of $\Lambda : \Gamma \to \mathbb{R}^{m \times n}$ is a binary matrix $S \in \{0,1\}^{m \times n}$ such that*

$$S_{i,j} = 1 \iff \exists \gamma \in \Gamma, \Lambda_{i,j}(\gamma) \neq 0.$$

An other way to phrase this is to say that the sparsity pattern of $\Lambda$ is the sparsest binary matrix $S$ such that $\Lambda(\Gamma) \subset \mathbb{R}^{m \times n}_S$.

We are now ready to present the lemmas that will be central to the main theorems of this work.

**Lemma 14** (Lachapelle et al. [2022]). *Let $S, S' \in \{0,1\}^{m \times m}$ and let $(A^{(i,j)})_{(i,j) \in S}$ be a basis of $\mathbb{R}^{m \times m}_S$. Let $L$ be a real $m \times m$ matrix. Then*

$$\forall (i,j) \in S, \ L^\top A^{(i,j)} L \in \mathbb{R}^{m \times m}_{S'} \iff \forall (i,j) \in S, \ (L_{i,\cdot})^\top L_{j,\cdot} \in \mathbb{R}^{m \times m}_{S'}. \tag{27}$$

*Proof.* We start with direction " $\implies$ ". Choose $(i_0, j_0) \in S$. Since $e_{i_0} e_{j_0}^\top \in \mathbb{R}^{m \times m}_S$ (where $e_i$ denotes the vector with a 1 at entry $i$ and 0 elsewhere) and the matrices $A^{(i,j)}$ form a basis of $\mathbb{R}^{m \times m}_S$, we can write $e_{i_0} e_{j_0}^\top = \sum_{(i,j) \in S} \alpha_{i,j} A^{(i,j)}$ for some coefficients $\alpha_{i,j}$. Thus

$$(L_{i_0,\cdot})^\top L_{j_0,\cdot} = L^\top e_{i_0} e_{j_0}^\top L \tag{28}$$

$$= L^\top \left( \sum_{(i,j) \in S} \alpha_{i,j} A^{(i,j)} \right) L \tag{29}$$

$$= \sum_{(i,j) \in S} \alpha_{i,j} L^\top A^{(i,j)} L \in \mathbb{R}^{m \times m}_{S'}, \tag{30}$$

where the final "$\in$" holds because each element of the sum is in $\mathbb{R}^{m \times m}_{S'}$.

We now show the reverse direction " $\impliedby$ ". Let $A \in \mathbb{R}^{m \times m}_S$. We can write

$$A = \sum_{(i,j) \in S} A_{i,j} e_i e_j^\top \tag{31}$$

$$L^\top A L = \sum_{(i,j) \in S} A_{i,j} L^\top e_i e_j^\top L = \sum_{(i,j) \in S} A_{i,j} (L_{i,\cdot})^\top L_{j,\cdot} \in \mathbb{R}^{m \times m}_{S'}, \tag{32}$$

where the last "$\in$" hold because every term in the sum is in $\mathbb{R}^{m \times m}_{S'}$. $\qquad\square$

**Lemma 15** (Lachapelle et al. [2022]). *Let $s, s' \in \{0,1\}^m$ and $(a^{(i)})_{i \in s}$ be a basis of $\mathbb{R}^m_s$. Let $L$ be a real $m \times m$ matrix. Then*

$$\forall i \in s, \ L^\top a^{(i)} \in \mathbb{R}^m_{s'} \iff \forall i \in s, \ (L_{i,\cdot})^\top \in \mathbb{R}^m_{s'}. \tag{33}$$

*Proof.* We start with "$\implies$". Choose $i_0 \in s$. We can write the one-hot vector $e_{i_0}$ as $\sum_{i \in s} \alpha_i a^{(i)}$ for some coefficients $\alpha_i$ (since $(a^{(i)})_{i \in s}$ forms a basis). Thus

$$(L_{i_0,\cdot})^\top = L^\top e_{i_0} = L^\top \sum_{i \in s} \alpha_i a^{(i)} = \sum_{i \in s} \alpha_i L^\top a^{(i)} \in \mathbb{R}^m_{s'}, \tag{34}$$

where the final "$\in$" holds because each element of the sum is in $\mathbb{R}^m_{s'}$.

We now show "$\impliedby$". Let $a \in \mathbb{R}^m_s$. We can write

$$a = \sum_{i \in s} a_i e_i \tag{35}$$

$$L^\top a = \sum_{i \in s} a_i L^\top e_i = \sum_{i \in s} a_i (L_{i,\cdot})^\top \in \mathbb{R}^m_{s'}, \tag{36}$$

where the last "$\in$" holds because all terms in the sum are in $\mathbb{R}^m_{s'}$. $\qquad\square$

The following simple Lemma will be useful throughout this section. The argument is taken from Lachapelle et al. [2022].

**Lemma 16** (Sparsity pattern of an invertible matrix contains a permutation). *Let $L \in \mathbb{R}^{m \times m}$ be an invertible matrix. Then, there exists a permutation $\sigma$ such that $L_{i,\sigma(i)} \neq 0$ for all $i$.*

*Proof.* Since the matrix $L$ is invertible, its determinant is non-zero, i.e.

$$\det(L) := \sum_{\sigma \in \mathfrak{S}_m} \mathrm{sign}(\sigma) \prod_{i=1}^m L_{i,\sigma(i)} \neq 0, \tag{37}$$

where $\mathfrak{S}_m$ is the set of $m$-permutations. This equation implies that at least one term of the sum is non-zero, meaning

$$\exists \sigma \in \mathfrak{S}_m, \forall i \leq m, L_{i,\sigma(i)} \neq 0. \tag{38}$$

$\qquad\square$

The exact form of the ground-truth graph $G$ will force some of the entries of the matrix $L$, which relates the ground-truth and the learned representations, to be zero. Understanding which entries of $L$ are zero is very important to understand *qualitatively* how disentangled the learned representation is expected to be. We now recall the notion of $S$-consistency (introduced in the main text) which will be crucial to precisely relate the form of the ground-truth graph $G$ to the sparsity pattern of $L$ via the consistency equivalence relation (Def. 7) in Thm. 8. Note that it is reformulated with the notation introduce in this appendix.

**Definition 6** ($S$-consistency). *Given a binary matrix $S \in \{0,1\}^{m \times n}$, a matrix $C \in \mathbb{R}^{m \times m}$ is $S$-consistent if*

$$C \in \mathbb{R}^{m \times m}_{[\mathbb{1} - S(\mathbb{1}-S)^\top]^+},$$

*where $[\cdot]^+ := \max\{0, \cdot\}$ and $\mathbb{1}$ is a matrix filled with ones (assuming implicitly its correct size).*

The following characterization of $S$-consistency will be useful later on to prove Lemma 18 & 19, to give an intuitive interpretation of $S$-consistency (Sec. A.2.8) and to relate $S$-consistency to the graphical criterion introduced by Lachapelle et al. [2022] (Sec. A.2.7).

**Lemma 17** (Characterizing $S$-consistency). *Let $C \in \mathbb{R}^{m \times m}$ and $S \in \{0,1\}^{m \times n}$. The following statements are equivalent.*

1. *$C$ is $S$-consistent (Def. 6);*
2. *$\forall i, (C_{i,\cdot})^\top \in \mathbb{R}^m_{\bigcap_{k \in S_{i,\cdot}} S_{\cdot,k}}$;*
3. *$\forall j, C_{\cdot,j} \in \mathbb{R}^m_{\bigcap_{k \in S^c_{j,\cdot}} S^c_{\cdot,k}}.$*

*Proof.* We proceed by showing how both the second and third statements are equivalent to the first one. Choose arbitrary $i$ and $j$.

$$[\mathbb{1} - S(\mathbb{1} - S)^\top]^+_{i,j} = 0 \iff 1 \leq S_{i,\cdot}(\mathbb{1} - S_{j,\cdot})^\top \tag{39}$$

$$\iff \exists k \text{ s.t. } S_{i,k} = 1 \text{ and } S_{j,k} = 0 \tag{40}$$

One can rephrase (40) as

$$\exists k \in S_{i,\cdot} \text{ s.t. } j \notin S_{\cdot,k} \iff j \notin \bigcap_{k \in S_{i,\cdot}} S_{\cdot,k}, \tag{41}$$

which proves the first and second statements are equivalent. One can also rephrase (40) as

$$\exists k \in S^c_{j,\cdot} \text{ s.t. } i \notin S^c_{\cdot,k} \iff i \notin \bigcap_{k \in S^c_{j,\cdot}} S^c_{\cdot,k}, \tag{42}$$

which proves the first and third statements are equivalent. $\qquad\square$

Later in Sec. A.2.3, we show that the set of invertible and $S$-consistent matrices form a group under matrix multiplication, i.e. that it is closed under matrix multiplication and inversion. This will be crucial to show that the relation $\sim_{\text{con}}$ (Def. 7) is an equivalence relation (Sec. A.2.4).

We are now ready to show the central lemmas that can be directly applied to easily prove the main theorem of this work, Thm. 8. Note that Lemmas 18 & 19 can be thought of as generalizations of Lemmas 17 & 18 from Lachapelle et al. [2022], respectively. The difference is that we do not assume anything about the specific form of $S$, which yields a different (sometime weaker) conclusion.

**Lemma 18** ($L^\top\Lambda(\cdot)L$ *sparse implies* $L$ *sparse*). *Let* $\Lambda : \Gamma \to \mathbb{R}^{m \times m}$ *with sparsity pattern* $S$ *(Def. 13). Let* $L \in \mathbb{R}^{m \times m}$ *be an invertible matrix and* $\hat{S}$ *be the sparsity pattern of* $\hat{\Lambda}(.) := L^\top\Lambda(\cdot)L$. *Let* $\sigma$ *be a permutation such that for all* $i$, $L_{i,\sigma(i)} \neq 0$ *(Lemma 16) and let* $P$ *be its associated permutation matrix, i.e.* $Pe_i = e_{\sigma(i)}$ *for all* $i$. *Assume that*

1. *[**Sufficient Variability**]* $\text{span}(\Lambda(\Gamma)) = \mathbb{R}^{m \times m}_S$.

*Then* $S \subset P^\top\hat{S}P$. *Further assume that*

2. *[**Sparsity**]* $||\hat{S}||_0 \leq ||S||_0$.

*Then* $S = P^\top\hat{S}P$ *and* $L = CP^\top$ *where* $C$ *is* $S$-*consistent and* $S^\top$-*consistent.*

*Proof.* We separate the proof in four steps. The first step leverages the Assumption 1 and Lemma 14 to show that $L$ must contain "many" zeros. The second step leverages the invertibility of $L$ to show that $PSP^\top \subset \hat{S}$. The third step uses Assumption 2 to establish $PSP^\top = \hat{S}$ and the fourth step concludes that $L = CP^\top$ where $C$ is both $S$-consistent and $S^\top$-consistent.

**Step 1:** By Assumption 1, there exists $(\gamma_{i,j})_{(i,j)\in S}$ such that $(\Lambda(\gamma_{i,j}))_{(i,j)\in S}$ spans $\mathbb{R}^{m \times m}_S$. Moreover, by the definition of $\hat{S}$ as sparsity pattern of $L^\top\Lambda(.)L$ (Definition 13), we have for all $(i,j) \in S$

$$L^\top\Lambda(\gamma_{i,j})L \in \mathbb{R}^{m \times m}_{\hat{S}}. \tag{43}$$

Then, by Lemma 14, we must have

$$\forall (i,j) \in S, \ (L_{i,\cdot})^\top L_{j,\cdot} \in \mathbb{R}^{m \times m}_{\hat{S}}. \tag{44}$$

**Step 2:** Since $\forall i, L_{i,\sigma(i)} \neq 0$, (44) implies that for all $(i,j) \in S$,

$$(\sigma(i), \sigma(j)) \in \hat{S}, \tag{45}$$

which, in other words, means that

$$PSP^\top \subset \hat{S}. \tag{46}$$

$$\tag{47}$$

This proves the first claim of the theorem.

**Step 3:** By Assumption 2, $||\hat{S}||_0 \leq ||S||_0 = ||PSP^\top||_0$, we must have that

$$PSP^\top = \hat{S}\,, \tag{48}$$

which proves the second statement of the theorem.

**Step 4:** We notice that, since $L_{j,\sigma(j)} \neq 0$, (44) implies

$$\forall i,\ \forall j \in S_{i,\cdot},\ (L_{i,\cdot})^\top \in \mathbb{R}^m_{\hat{S}_{\cdot,\sigma(j)}} \tag{49}$$

$$\text{and } \forall j,\ \forall i \in S_{\cdot,j},\ (L_{j,\cdot})^\top \in \mathbb{R}^m_{\hat{S}_{\sigma(i),\cdot}}\,, \tag{50}$$

We interchange indices $i$ and $j$ in the second equation above (this is purely a change of notation), which yields

$$\forall i,\ \forall j \in S_{\cdot,i},\ (L_{i,\cdot})^\top \in \mathbb{R}^m_{\hat{S}_{\sigma(j),\cdot}}\,, \tag{51}$$

Equations (49) & (51) can be rewritten as

$$\forall i,\ (L_{i,\cdot})^\top \in \bigcap_{j \in S_{i,\cdot}} \mathbb{R}^m_{\hat{S}_{\cdot,\sigma(j)}} = \mathbb{R}^m_{\bigcap_{j \in S_{i,\cdot}} \hat{S}_{\cdot,\sigma(j)}} \tag{52}$$

$$\forall i,\ (L_{i,\cdot})^\top \in \bigcap_{j \in S_{\cdot,i}} \mathbb{R}^m_{\hat{S}_{\sigma(j),\cdot}} = \mathbb{R}^m_{\bigcap_{j \in S_{\cdot,i}} \hat{S}_{\sigma(j),\cdot}} \tag{53}$$

Applying left multiplying both equations above by $P^\top$, we obtain

$$\forall i,\ ((LP)_{i,\cdot})^\top = P^\top (L_{i,\cdot})^\top \in P^\top \mathbb{R}^m_{\bigcap_{j \in S_{i,\cdot}} \hat{S}_{\cdot,\sigma(j)}} = \mathbb{R}^m_{\bigcap_{j \in S_{i,\cdot}} S_{\cdot,j}} \tag{54}$$

$$\forall i,\ ((LP)_{i,\cdot})^\top = P^\top (L_{i,\cdot})^\top \in P^\top \mathbb{R}^m_{\bigcap_{j \in S_{\cdot,i}} \hat{S}_{\sigma(j),\cdot}} = \mathbb{R}^m_{\bigcap_{j \in S_{\cdot,i}} S_{j,\cdot}} = \mathbb{R}^m_{\bigcap_{j \in (S^\top)_{i,\cdot}} (S^\top)_{\cdot,j}} \tag{55}$$

Let $C := LP$. Equations (54) & (55) imply that $C$ is $S$-consistent and $S^\top$-consistent, respectively (by Lemma 17). Since $L = CP^\top$, this completes the proof. $\square$

**Lemma 19** ($L^\top \Lambda(.)$ sparse implies $L$ sparse)**.** *Let* $\Lambda : \Gamma \to \mathbb{R}^{m \times n}$ *with sparsity pattern* $S$. *Let* $L \in \mathbb{R}^{m \times m}$ *be an invertible matrix and* $\hat{S}$ *be the sparsity pattern of* $\hat{\Lambda} := L^\top \Lambda$. *Let* $\sigma$ *be a permutation such that for all* $i$, $L_{i,\sigma(i)} \neq 0$ *(Lemma 16) and let* $P$ *be its associated permutation matrix, i.e.* $Pe_i = e_{\sigma(i)}$ *for all* $i$. *Assume that*

1. *[**Sufficient Variability**] For all* $j \in \{1, ..., n\}$, $span(\Lambda_{\cdot,j}(\Gamma)) = \mathbb{R}^m_{S_{\cdot,j}}$ .

*Then* $S \subset P^\top \hat{S}$. *Further assume that*

2. *[**Sparsity**]* $||\hat{S}||_0 \leq ||S||_0$ .

*Then* $S = P^\top \hat{S}$ *and* $L = CP^\top$ *where* $C$ *is* $S$-*consistent.*

*Proof.* We separate the proof in four steps. The first step leverages the Assumption 1 and Lemma 15 to show that $L$ must contain "many" zeros. The second step leverages the invertibility of $L$ to show that $PS \subset \hat{S}$. The third step uses Assumption 2 to show this inclusion is in fact an equality and the fourth step concludes that $L$ can be written as an $S$-consistent matrix times $P^\top$.

**Step 1:** Fix $j \in \{1, ..., n\}$. By Assumption 1, there exists $(\gamma_i)_{i \in S_{\cdot,j}}$ such that $(\Lambda_{\cdot,j}(\gamma_i))_{i \in S_{\cdot,j}}$ spans $\mathbb{R}^m_{S_{\cdot,j}}$. Moreover, by the definition of $\hat{S}$ as sparsity pattern of $L^\top \Lambda(.)$ (Definition 13), we have for all $i \in S_{\cdot,j}$

$$L^\top \Lambda_{\cdot,j}(\gamma_i) \in \mathbb{R}^m_{\hat{S}_{\cdot,j}}\,. \tag{56}$$

By Lemma 15, we must have

$$\forall\, i \in S_{\cdot,j},\ (L_{i,\cdot})^\top \in \mathbb{R}^m_{\hat{S}_{\cdot,j}}\,. \tag{57}$$

Since $j$ was arbitrary, this holds for all $j$, which allows us to rewrite as

$$\forall (i,j) \in S,\ (L_{i,\cdot})^\top \in \mathbb{R}^m_{\hat{S}_{\cdot,j}}\,. \tag{58}$$

**Step 2:** Since $L_{i,\sigma(i)} \neq 0$ for all $i$, (58) implies that

$$\forall (i,j) \in S,\ (\sigma(i),j) \in \hat{S}_{\cdot,j}\,, \tag{59}$$

which can be rephrased as

$$PS \subset \hat{S}\,. \tag{60}$$

This proves the first statement of the theorem.

**Step 3:** By Assumption 2, $||\hat{S}||_0 \leq ||S||_0 = ||PS||_0$, so the inclusion (60) is actually an equality

$$PS = \hat{S}\,, \tag{61}$$

which proves the second statement.

**Step 4:** We notice that (58) can be rewritten as

$$\forall i, \forall j \in S_{i,\cdot},\ (L_{i,\cdot})^\top \in \mathbb{R}^m_{\hat{S}_{\cdot,j}}\,, \tag{62}$$

which is equivalent to

$$\forall i,\ (L_{i,\cdot})^\top \in \bigcap_{j \in S_{i,\cdot}} \mathbb{R}^m_{\hat{S}_{\cdot,j}} = \mathbb{R}^m_{\bigcap_{j \in S_{i,\cdot}} \hat{S}_{\cdot,j}}\,. \tag{63}$$

We apply $P^\top$ on both sides of the above line and get

$$\forall i,\ ((LP)_{i,\cdot})^\top = P^\top (L_{i,\cdot})^\top \in P^\top \mathbb{R}^m_{\bigcap_{j \in S_{i,\cdot}} \hat{S}_{\cdot,j}} = \mathbb{R}^m_{\bigcap_{j \in S_{i,\cdot}} S_{\cdot,j}}\,, \tag{64}$$

which means, by Lemma 17, that $LP$ is $S$-consistent. By defining $C := LP$, we have that $L = CP^\top$, which is what we wanted to prove. $\qquad\square$

### A.2.3 Invertible $S$-consistent matrices form a group under matrix multiplication

The following theorem shows that, perhaps surprisingly, the set of invertible $S$-consistent matrices forms a *group* under matrix multiplication, i.e. that the set is closed under multiplication and inversion. This will be very useful to show that the consistence relation over models, $\sim_{\mathrm{con}}$ (Def. 7), is an equivalence relation. The proof can be safely skipped at first read.

**Theorem 20.** *Let $S \in \{0,1\}^{m \times n}$.*

1. *The identity matrix $I$ is $S$-consistent;*

2. *For any invertible $S$-consistent matrices $C$ and $C'$, the matrix product $CC'$ is also $S$-consistent;*

3. *For any invertible $S$-consistent matrix $C$, $C^{-1}$ is also $S$-consistent.*

*In other words, the set of invertible matrices that are $S$-consistent forms a group under matrix multiplication.*

*Proof.* First, let $i \leq m$. Notice that $[\mathbb{1} - S(\mathbb{1} - S)^\top]^+_{i,i} = [1 - S_{i,\cdot}(\mathbb{1}_{i,\cdot} - S_{i,\cdot})^\top]^+ = 1$. Thus, $I$ is $S$-consistent.

Second, we show closure under matrix multiplication. Let $i, j$ such that $[\mathbb{1}-S(\mathbb{1}-S)^\top]^+_{i,j} = 0$. Consider $(CC')_{i,j} = C_{i,\cdot}C'_{\cdot,j}$. By Lemma 17, we have that

$$(C_{i,\cdot})^\top \in \mathbb{R}^m_{\bigcap_{k \in S_{i,\cdot}} S_{\cdot,k}} \tag{65}$$

$$C'_{\cdot,j} \in \mathbb{R}^m_{\bigcap_{k \in S^c_{j,\cdot}} S^c_{\cdot,k}} \tag{66}$$

Notice that if the intersection $\left(\bigcap_{k \in S_{i,\cdot}} S_{\cdot,k}\right) \cap \left(\bigcap_{k \in S^c_{j,\cdot}} S^c_{\cdot,k}\right)$ is empty, the dot product $C_{i,\cdot}C'_{\cdot,j}$ is zero and the second statement of this theorem holds. By (40) from the proof of Lemma 17, there exists a $k$ such that $k \in S_{i,\cdot}$ and $k \in S^c_{j,\cdot}$, and, since $S_{\cdot,k} \cap S^c_{\cdot,k} = \emptyset$, the initial intersection is itself empty.

Third, we show that the inverse $C^{-1}$ is also $S$-consistent. Notice that, since $C$ is invertible, there exists a sequence of *elementary row operations* that will transform $C$ into the identity. This process is sometimes called Gaussian elimination or Gauss-Jordan elimination. The elementary row operations are (i) swapping two rows, (ii) multiplying a row by a nonzero number, and (iii) adding a multiple of one row to another. These three elementary operation can be performed by left multiplying by an *elementary matrix*, which have the following forms:

(i) Swapping two rows:

$$\begin{bmatrix} 1 & & & & & & \\ & \ddots & & & & & \\ & & 0 & & 1 & & \\ & & & \ddots & & & \\ & & 1 & & 0 & & \\ & & & & & \ddots & \\ & & & & & & 1 \end{bmatrix} ; \tag{67}$$

(ii) Multiplying a row by a nonzero number:

$$\begin{bmatrix} 1 & & & & & \\ & \ddots & & & & \\ & & 1 & & & \\ & & & \alpha & & \\ & & & & 1 & \\ & & & & & \ddots & \\ & & & & & & 1 \end{bmatrix} ; \tag{68}$$

(iii) Adding a multiple of a row to another:

$$\begin{bmatrix} 1 & & & & & \\ & \ddots & & & & \\ & & 1 & & & \\ & & & \ddots & & \\ & & \alpha & & 1 & \\ & & & & & \ddots & \\ & & & & & & 1 \end{bmatrix} . \tag{69}$$

We will show that it is possible to transform $C$ into the identity *by using only elementary matrices that are themselves S-consistent*, i.e. that there exists a sequence of $S$-consistent elementary matrices $E_1, ..., E_p$, such that $E_p...E_2 E_1 C = I$. Since this implies $C^{-1} = E_p...E_2 E_1$ and all elementary matrices are $S$-consistent, $C^{-1}$ is also $S$-consistent (using closure under multiplication shown above).

We now construct the sequence of $E_i$ using standard Gaussian elimination. Start by initializing $M := C$. Throughout the algorithm, $M$ will be gradually transformed by elementary operations that are $S$-consistent (and invertible), thus $M$ will

remain $S$-consistent (and invertible). We consider every column $j = 1, ..., m$ from left to right. If $M_{j,j} = 0$, we will show that rows $j, ..., m$ can be permuted to obtain $M_{j,j} \neq 0$ using an $S$-consistent permutation, but we delay this technical step to the end of the proof to avoid breaking the flow of the exposition. For now, assume $M_{j,j} \neq 0$. Rescale row $j$ so that $M_{j,j} = 1$ using matrix of the form (68), which is $S$-consistent. Then, put zeroes below $M_{j,j}$ by adding a multiple of row $j$ to each row $i > j$ such that $M_{i,j} \neq 0$. Each of these operations corresponds to an elementary matrix of the form (69) where the nonzero entry below the diagonal is at position $(i, j)$. Since $M$ is $S$-consistent and $M_{i,j} \neq 0$, these elementary matrices must also be $S$-consistent. Once every element below $M_{j,j}$ are zero go to the next column. Do that for all columns.

At this point, $M$ is upper triangular with a diagonal filled with ones. We must now remove every nonzero elements above the diagonal by a process similar to what we just did. Start with column $j = n$ up to $j = 1$, from left to right. To remove every nonzero elements above $M_{j,j}$, we can add a multiple of row $j$ to the rows $i < j$ that have $M_{i,j} \neq 0$. This is equivalent to multiplying $M$ by an elementary matrix of the form (69) with its off diagonal nonzero entry by at position $(i, j)$. Again, since $M_{i,j} \neq 0$ and $M$ is $S$-consistent, this elementary matrix must also be $S$-consistent. Once all elements above $M_{j,j}$ are zeros, go to the next column and repeat for every columns until column $j = 1$ is reached.

At this point, $M = I$, which is what we wanted to show.

We now have to show what to do when $M_{j,j} \neq 0$. We know that $M$ has the following form

$$M = \begin{bmatrix} U & A \\ 0 & B \end{bmatrix}, \tag{70}$$

where $U \in \mathbb{R}^{(j-1) \times (j-1)}$ is an upper triangular matrix with only ones on its diagonal and $B$ is a square matrix with $B_{1,1} = 0$. Since $M$ is invertible, $B$ is invertible too (otherwise, $\det(M) = \det(U)\det(B) = 0$). Thus, by Lemma 16, there exists a permutation $\sigma$ such that for all $i$, $B_{\sigma(i),i} \neq 0$. Consider its corresponding permutation matrix $P := [e_{\sigma(1)} \cdots e_{\sigma(m-j+1)}]$. Notice that the matrix

$$\begin{bmatrix} I_{j-1} & 0 \\ 0 & P \end{bmatrix} \tag{71}$$

is $S$-consistent, since otherwise $M$ is not. We know that the *cyclic group* $\{P^k \mid k \in \mathbb{Z}\}$ forms a subgroup of the group of permutations, and thus has finite order. Thus, there exists $\ell \in \mathbb{N}$ such that $P^\ell = I$, and thus $P^{-1} = P^{\ell-1}$ [Artin, 2013, Section 2.4]. Recall $P^{-1} = P^\top$ since $P$ is a permutation. This means

$$\begin{bmatrix} I_{j-1} & 0 \\ 0 & P^\top \end{bmatrix} \tag{72}$$

is $S$-consistent, since it is a product of $S$-consistent matrices. Notice how $(P^\top B)_{i,i} = e_{\sigma(i)}^\top B_{\cdot,i} = B_{\sigma(i),i} \neq 0$. In particular $(P^\top B)_{1,1} \neq 0$. We can thus update $M$ by applying matrix (72) to it to get a nonzero entry at $(j, j)$:

$$M \leftarrow \underbrace{\begin{bmatrix} U & A \\ 0 & P^\top B \end{bmatrix}}_{\substack{\text{Still } S\text{-consistent} \\ + \text{ entry } (j,j) \text{ nonzero}}} = \underbrace{\begin{bmatrix} I_{j-1} & 0 \\ 0 & P^\top \end{bmatrix}}_{S\text{-consistent}} \underbrace{\begin{bmatrix} U & A \\ 0 & B \end{bmatrix}}_{M}, \tag{73}$$

which completes the proof. $\qquad\square$

### A.2.4 The consistency relation (Def. 7) is an equivalence relation

We start by showing a fact that will be useful to show that $\sim_{\text{con}}$ is an equivalence relation.

**Lemma 21.** *Let* $S \in \{0, 1\}^{m \times n}$.

1. *A matrix* $C \in \mathbb{R}^{m \times m}$ *is $S$-consistent if and only if $C$ is $SP$-consistent, where $P$ is an $n \times n$ permutation matrix.*

2. *A matrix* $C \in \mathbb{R}^{m \times m}$ *is $S$-consistent if and only if $PCP^\top$ is $PS$-consistent, where $P$ is a $m \times m$ permutation matrix.*

3. *When $m = n$, a matrix* $C \in \mathbb{R}^{m \times m}$ *is $S$-consistent if and only if $PCP^\top$ is $PSP^\top$-consistent, where $P$ is a $m \times m$ permutation matrix.*

*Proof.* To show the first statement, we simply have to notice that

$$[\mathbb{1} - SP(\mathbb{1} - SP)^\top]^+ = [\mathbb{1} - SPP^\top(\mathbb{1} - S)^\top]^+ \tag{74}$$

$$= [\mathbb{1} - S(\mathbb{1} - S)^\top]^+ \tag{75}$$

To show the second statement, we start with

$$C \in \mathbb{R}^{m \times m}_{[\mathbb{1} - S(\mathbb{1}-S)^\top]^+} \tag{76}$$

$$\iff PCP^\top \in P\mathbb{R}^{m \times m}_{[\mathbb{1} - S(\mathbb{1}-S)^\top]^+} P^\top \tag{77}$$

$$= \mathbb{R}^{m \times m}_{P[\mathbb{1} - S(\mathbb{1}-S)^\top]^+ P^\top} \tag{78}$$

$$= \mathbb{R}^{m \times m}_{[\mathbb{1} - PS(\mathbb{1}-PS)^\top]^+} . \tag{79}$$

The third statement, is a combination of the first two. $\qquad\square$

**Proposition 22.** *The consistency relation, $\sim_{\mathrm{con}}$ (Def. 7), is an equivalence relation.*

*Proof.* First, recall the fact that an intersection of subgroups is a subgroup. This means that, the set of invertible matrices that are $G^z$-consistent, $(G^z)^\top$-consistent and $G^a$-consistent is a group, and thus is closed under matrix multiplication and inversion.

**Reflexivity.** It is easy to see that $\theta \sim_{\mathrm{con}} \theta$, by simply setting $L := I$.

**Symmetry.** Assume $\theta \sim_{\mathrm{con}} \tilde{\theta}$. Hence, we have $G^z = P^\top \tilde{G}^z P$ and $G^a = P^\top \tilde{G}^a$ as well as

$$\mathbf{T}(\mathbf{f}^{-1}(x)) = CP^\top \mathbf{T}(\tilde{\mathbf{f}}^{-1}(x)) + b \,, \text{ and} \tag{80}$$

$$PC^\top \boldsymbol{\lambda}(\mathbf{f}^{-1}(x^{<t}), a^{<t}) + c = \tilde{\boldsymbol{\lambda}}(\tilde{\mathbf{f}}^{-1}(x^{<t}), a^{<t}) \,. \tag{81}$$

where the matrix $C$ is $G^z$-consistent, $(G^z)^\top$-consistent and $G^a$-consistent.

In order to show symmetry, we just need to show that the inverse of $CP^\top$ can be written as $\tilde{C}\tilde{P}^\top$ where $\tilde{P}$ is some permutation and $\tilde{C}$ is $\tilde{G}^z$-consistent, $(\tilde{G}^z)^\top$-consistent and $\tilde{G}^a$-consistent. Notice that $(CP^\top)^{-1} = PC^{-1}$ and that $C^{-1}$ is consistent to $G^z$, $(G^z)^\top$ and $G^a$ by closure under inversion. Thus, by Lemma 21, we have that $\tilde{C} := PC^{-1}P^\top$ is $\tilde{G}^z$-consistent, $(\tilde{G}^z)^\top$-consistent, $\tilde{G}^a$-consistent. Hence

$$(CP^\top)^{-1} = PC^{-1} = \underbrace{PC^{-1}P^\top}_{\tilde{C}} \underbrace{P}_{\tilde{P}^\top} = \tilde{C}\tilde{P}^\top \,. \tag{82}$$

**Transitivity.** Suppose $\theta \sim_{\mathrm{con}} \tilde{\theta}$ and $\tilde{\theta} \sim_{\mathrm{con}} \hat{\theta}$. This means

$$G^z = P_1^\top \tilde{G}^z P_1 \text{ and } G^a = P_1^\top \tilde{G}^a \,, \tag{83}$$

$$\mathbf{T}(\mathbf{f}^{-1}(x)) = C_1 P_1^\top \mathbf{T}(\tilde{\mathbf{f}}^{-1}(x)) + b_1 \,, \text{ and} \tag{84}$$

$$P_1 C_1^\top \boldsymbol{\lambda}(\mathbf{f}^{-1}(x^{<t}), a^{<t}) + c_1 = \tilde{\boldsymbol{\lambda}}(\tilde{\mathbf{f}}^{-1}(x^{<t}), a^{<t}) \,, \tag{85}$$

where $C_1$ is consistent to $G^z$, $(G^z)^\top$ and $G^a$; and

$$\tilde{G}^z = P_2^\top \hat{G}^z P_2 \text{ and } \tilde{G}^a = P_2^\top \hat{G}^a \,, \tag{86}$$

$$\mathbf{T}(\tilde{\mathbf{f}}^{-1}(x)) = C_2 P_2^\top \mathbf{T}(\hat{\mathbf{f}}^{-1}(x)) + b_2 \,, \text{ and} \tag{87}$$

$$P_2 C_2^\top \tilde{\boldsymbol{\lambda}}(\mathbf{f}^{-1}(x^{<t}), a^{<t}) + c_2 = \hat{\boldsymbol{\lambda}}(\hat{\mathbf{f}}^{-1}(x^{<t}), a^{<t}) \,, \tag{88}$$

where $C_2$ is consistent to $\tilde{G}^z$, $(\tilde{G}^z)^\top$ and $\tilde{G}^a$.

To show that $\theta \sim_{\mathrm{con}} \hat{\theta}$, we first combine (83) with (86) to get

$$G^z = \underbrace{P_1^\top P_2^\top}_{P^\top} \hat{G}^z \underbrace{P_2 P_1}_{P} \text{ and } G^a = \underbrace{P_1^\top P_2^\top}_{P^\top} \hat{G}^a \,. \tag{89}$$

Moreover, we can combine (84) with (87) to get

$$\mathbf{T}(\mathbf{f}^{-1}(x)) = C_1 P_1^\top (C_2 P_2^\top \mathbf{T}(\hat{\mathbf{f}}^{-1}(x)) + b_2) + b_1 \tag{90}$$

$$= C_1 P_1^\top C_2 P_2^\top \mathbf{T}(\hat{\mathbf{f}}^{-1}(x)) + (C_1 P_1^\top b_2 + b_1), \tag{91}$$

and the same can be done for (85) and (88). We must now show that $C_1 P_1^\top C_2 P_2^\top = C P^\top$ where $C$ is some matrix consistent to $G^z$, $(G^z)^\top$ and $G^a$ (Def. 6). Notice that

$$C_1 P_1^\top C_2 P_2^\top = C_1 P_1^\top P_2^\top \underbrace{(P_2 C_2 P_2^\top)}_{\hat{C}}, \tag{92}$$

where $\hat{C} := P_2 C_2 P_2^\top$ is consistent to $\hat{G}^z$, $(\hat{G}^z)^\top$ and $\hat{G}^a$, by Lemma 21. We can further write

$$C_1 P_1^\top C_2 P_2^\top = C_1 P_1^\top P_2^\top \hat{C} \tag{93}$$

$$= C_1 \underbrace{(P_1^\top P_2^\top \tilde{C} P_2 P_1)}_{C'} \underbrace{P_1^\top P_2^\top}_{P^\top}, \tag{94}$$

where $C' := P_1^\top P_2^\top \tilde{C} P_2 P_1$ is consistent to $G^z$, $(G^z)^\top$ and $G^a$, by Lemma 21 and (89). Since $C_1$ is also consistent to $G^z$, $(G^z)^\top$ and $G^a$, the product $C := C_1 C'$ also is, because of closure under multiplication (Thm. 20). This concludes the proof that $\theta \sim_{\mathrm{con}} \hat{\theta}$. $\qquad\square$

### A.2.5   Proof of Theorem 8

Finally, we can prove Thm. 8. Note that its proof reuses many arguments initially introduced by Lachapelle et al. [2022]. In fact, the statement of Thm. 8 is identical to Thm. 5 of Lachapelle et al. [2022] except for (i) the absence of the graphical criterion (Def. 23) and (ii) the conclusion, which is $\theta \sim_{\mathrm{con}} \hat{\theta}$ instead of $\theta \sim_{\mathrm{perm}} \hat{\theta}$. App. A.2.7 shows how Thm. 8 can be seen as a generalization of Thm. 5 from Lachapelle et al. [2022].

**Theorem 8** (Disentanglement via mechanism sparsity). *Suppose we have two models as described in Sec. 2.1 with parameters $\theta = (\mathbf{f}, \boldsymbol{\lambda}, G)$ and $\hat{\theta} = (\hat{\mathbf{f}}, \hat{\boldsymbol{\lambda}}, \hat{G})$ representing the same distribution, i.e. $\mathbb{P}_{X^{\leq T}|a;\theta} = \mathbb{P}_{X^{\leq T}|a;\hat{\theta}}$ for all $a \in \mathcal{A}^T$. Suppose the assumptions of Thm. 4 hold and that*

1. *The sufficient statistic $\mathbf{T}$ is $d_z$-dimensional ($k = 1$) and is a diffeomorphism from $\mathcal{Z}$ to $\mathbf{T}(\mathcal{Z})$.*

2. *[**Sufficient time-variability**] There exist $\{(z_{(p)}, a_{(p)}, \tau_{(p)})\}_{p=1}^{||G^z||_0}$ belonging to their respective support such that*

$$\mathrm{span}\left\{ D_z^{\tau(p)} \boldsymbol{\lambda}(z_{(p)}, a_{(p)}) D_z \mathbf{T}(z_{(p)}^{\tau(p)})^{-1} \right\}_{p=1}^{||G^z||_0} = \mathbb{R}_{G^z}^{d_z \times d_z},$$

   *where $D_z^{\tau(p)}$ and $D_z$ are the Jacobian operators with respect to $z^{\tau(p)}$ and $z$, respectively.*

*Then, there exists a permutation matrix $P$ such that $G^z \subset P^\top \hat{G}^z P$. Further assume that*

3. *[**Sufficient action-variability**] For all $\ell \in \{1, ..., d_a\}$, there exist $\{(z_{(p)}, a_{(p)}, \epsilon_{(p)}, \tau_{(p)})\}_{p=1}^{|\mathbf{Ch}_\ell^a|}$ belonging to their respective support such that*

$$\mathrm{span}\left\{ \Delta_\ell^{\tau(p)} \boldsymbol{\lambda}(z_{(p)}, a_{(p)}, \epsilon_{(p)}) \right\}_{p=1}^{|\mathbf{Ch}_\ell^a|} = \mathbb{R}_{\mathbf{Ch}_\ell^a}^{d_z},$$

   *where $\mathbf{Ch}_\ell^a$ is the set of children of $a_\ell$ and $\Delta_\ell^\tau \boldsymbol{\lambda}(z^{<t}, a^{<t}, \epsilon)$ is a partial difference defined by*

$$\Delta_\ell^\tau \boldsymbol{\lambda}(z^{<t}, a^{<t}, \epsilon) := \boldsymbol{\lambda}(z^{<t}, a^{<t} + \epsilon E_{\ell,\tau}) - \boldsymbol{\lambda}(z^{<t}, a^{<t}), \tag{95}$$

   *where $\epsilon \in \mathbb{R}$ and $E_{\ell,\tau} \in \mathbb{R}^{d_a \times t}$ is the one-hot matrix with the entry $(\ell, \tau)$ set to one. Thus, (95) is the discrete analog of a partial derivative w.r.t. $a_\ell^\tau$.*

*Then $G^a \subset P^\top \hat{G}^a$. Further assume that*

4. *[Sparsity]* $||\hat{G}||_0 \leq ||G||_0$.

*Then, $\hat{\theta}$ is consistent with $\theta$, i.e. $\theta \sim_{\mathrm{con}} \hat{\theta}$ (Def. 7).*

*Proof.* First of all, since the assumptions of Thm. 4 hold, we have that $\theta$ and $\hat{\theta}$ are linearly equivalent. Since $k = 1$ (assumption 1), we can apply Lemma 10 to obtain the following equations:

$$L^\top \underbrace{D_z^\tau \boldsymbol{\lambda}(z^{<t}, a^{<t}) D\mathbf{T}(z^\tau)^{-1}}_{\Lambda^{(1)}(\gamma)} L = \underbrace{D_z^\tau \hat{\boldsymbol{\lambda}}(\mathbf{v}(z^{<t}), a^{<t}) D\mathbf{T}(\mathbf{v}(z^\tau))^{-1}}_{\hat{\Lambda}^{(1)}(\gamma)}, \text{ and} \tag{96}$$

$$L^\top \underbrace{\Delta^\tau \boldsymbol{\lambda}(z^{<t}, a^{<t}, \vec{\epsilon})}_{\Lambda^{(2)}(\gamma)} = \underbrace{\Delta^\tau \hat{\boldsymbol{\lambda}}(\mathbf{v}(z^{<t}), a^{t-1}, \vec{\epsilon})}_{\hat{\Lambda}^{(2)}(\gamma)}, \tag{97}$$

where we use the labelling of Sec. A.2.1 with $\Lambda$ functions. Let us introduce $S^{(1)}, \hat{S}^{(1)}, S^{(2)}$ and $\hat{S}^{(2)}$, the sparsity patterns of $\Lambda^{(1)}, \hat{\Lambda}^{(1)}, \Lambda^{(2)}$ and $\hat{\Lambda}^{(2)}$, respectively. As was hinted at in Sec. A.2.1, the relationship between the sparsity patterns and the graphs is

$$S^{(1)} \subset G^z, \quad \hat{S}^{(1)} \subset \hat{G}^z, \tag{98}$$

$$S^{(2)} \subset G^a, \quad \hat{S}^{(2)} \subset \hat{G}^a. \tag{99}$$

Because of assumptions 2 & 3, we must have that

$$S^{(1)} = G^z, \tag{100}$$

$$S^{(2)} = G^a. \tag{101}$$

Notice how assumption 2 corresponds to assumption 1 of Lemma 18 and how assumption 3 corresponds to assumption 1 of Lemma 19. This means we can obtain the first conclusion of both Lemmas 18 & 19, i.e. that

$$S^{(1)} \subset P^\top \hat{S}^{(1)} P, \text{ and } S^{(2)} \subset P^\top \hat{S}^{(2)}, \tag{101}$$

which implies

$$||S^{(1)}||_0 \leq ||\hat{S}^{(1)}||_0 \text{ and } ||S^{(2)}||_0 \leq ||\hat{S}^{(2)}||_0 \tag{102}$$

All the above together with the sparsity assumption ($||\hat{G}||_0 \leq ||G||_0$) allows to write

$$||\hat{S}^{(1)}||_0 + ||\hat{S}^{(2)}||_0 \leq ||\hat{G}^z||_0 + ||\hat{S}^{(2)}||_0 \qquad \text{[By (98)]} \tag{103}$$

$$\leq ||\hat{G}^z||_0 + ||\hat{G}^a||_0 \qquad \text{[By (99)]} \tag{104}$$

$$= ||\hat{G}||_0 \tag{105}$$

$$\leq ||G||_0 \qquad \text{[By assumption 4 (Sparsity)]} \tag{106}$$

$$= ||G^z||_0 + ||G^a||_0 \tag{107}$$

$$= ||S^{(1)}||_0 + ||S^{(2)}||_0 \qquad \text{[By (100)]} \tag{108}$$

$$\leq ||\hat{S}^{(1)}||_0 + ||S^{(2)}||_0 \qquad \text{[By (102)]} \tag{109}$$

$$\leq ||\hat{S}^{(1)}||_0 + ||\hat{S}^{(2)}||_0 \qquad \text{[By (102)]}. \tag{110}$$

Since the l.h.s. of (103) equals the r.h.s. of (110), all the above inequalities are actually equalities. Hence we have

$$||\hat{S}^{(1)}||_0 = ||S^{(1)}||_0 \text{ and } ||\hat{S}^{(2)}||_0 = ||S^{(2)}||_0, \tag{111}$$

as well as

$$||\hat{S}^{(1)}||_0 = ||\hat{G}^z||_0 \text{ and } ||\hat{S}^{(2)}||_0 = ||\hat{G}^a||_0. \tag{112}$$

The latter, combined with the r.h.s. of (98) and (99), implies that

$$\hat{S}^{(1)} = \hat{G}^z \text{ and } \hat{S}^{(2)} = \hat{G}^a. \tag{113}$$

The equalities of (111) respectively implies the inequalities of the sparsity assumption of Lemmas 18 & 19, which allows us to obtain their second and most important conclusion i.e. that $S^{(1)} = P^\top \hat{S}^{(1)} P$, $S^{(2)} = P^\top \hat{S}^{(2)}$ and that $L = CP^\top$ where $C$ is $S^{(1)}$-consistent, $(S^{(1)})^\top$-consistent (Lemma 18) and $S^{(2)}$-consistent (Lemma 19). Notice that because $S^{(1)} = G^z$, $S^{(2)} = G^a$, $\hat{S}^{(1)} = \hat{G}^z$ and $\hat{S}^{(2)} = \hat{G}^a$, these are equivalent to what we wanted to show, i.e. that $\theta \sim_{\mathrm{con}} \hat{\theta}$. $\qquad\square$

### A.2.6 Understanding the sufficient variability assumptions of Thm. 8

To gain a better understanding of *sufficient time-variability* and *sufficient action-variability* assumptions of Thm. 8, we provide examples of transition functions $\boldsymbol{\lambda}$ that do not satisfy them. The synthetic datasets used in our experiments are examples of processes satisfying the sufficient variability assumption, their exact form can be found in App. B.1.

For the sake of simplicity, assume the latent variables are Gaussian with a variance fixed to one, which implies that $\mathbf{T}$ is the identity. Further assume that the system is Markovian, meaning $\boldsymbol{\lambda}(z^{<t}, a^{<t}) = \boldsymbol{\lambda}(z^{t-1}, a^{t-1})$. The sufficient time-variability thus reduces to: There exist $\{(z_{(p)}, a_{(p)})\}_{p=1}^{||G^z||_0}$ belonging to their respective support such that

$$\text{span}\left\{D_z\boldsymbol{\lambda}(z_{(p)}, a_{(p)})\right\}_{p=1}^{||G^z||_0} = \mathbb{R}_{G^z}^{d_z \times d_z}.$$

Now, assume $\boldsymbol{\lambda}(z^{t-1}, a^{t-1}) := W z^{t-1}$, with $W \in \mathbb{R}_{G^z}^{d_z \times d_z}$. This implies that $D_z\boldsymbol{\lambda}(z^{t-1}, a^{t-1}) = W$, which clearly means that the sufficient time-variability assumption is not satisfied. In this context, this assumption requires that $\boldsymbol{\lambda}$ is sufficiently nonlinear, in the sense that its Jacobian matrix varies sufficiently. We postulate that this assumption is a reasonable one, given how complex real world dynamics can be.

Similarly, assume that $\boldsymbol{\lambda}(z^{<t-1}, a^{t-1}) = W a^{t-1}$, with $W \in \mathbb{R}_{G^a}^{d_z \times d_a}$. We thus have that

$$\Delta_\ell \boldsymbol{\lambda}(z^{t-1}, a^{t-1}, \epsilon) := \boldsymbol{\lambda}(z^{t-1}, a^{t-1} + \epsilon e_\ell) - \boldsymbol{\lambda}(z^{t-1}, a^{t-1}) \tag{114}$$

$$= W(a^{t-1} + \epsilon e_\ell) - W a^{t-1} \tag{115}$$

$$= \epsilon W_{\cdot, \ell}. \tag{116}$$

Unless every $a_\ell$ has exactly one child, the sufficient action-variability assumption is violated, which, again, shows how linearity can cause problem.

### A.2.7 Connecting to the graphical criterion of Lachapelle et al. [2022]

We now clarify how the graphical criterion of Lachapelle et al. [2022], which guarantees complete disentanglement, is related to Thm. 8. Let us first recall what this criterion is about.

**Definition 23** (Graphical criterion of Lachapelle et al. [2022]). *A graph $G = [G^z\ G^a] \in \{0,1\}^{d_z \times (d_z + d_a)}$ satisfies the criterion of Lachapelle et al. [2022] if, for all $i \in \{1, ..., d_z\}$,*

$$\left(\bigcap_{j \in \mathbf{Ch}_i^z} \mathbf{Pa}_j^z\right) \cap \left(\bigcap_{j \in \mathbf{Pa}_i^z} \mathbf{Ch}_j^z\right) \cap \left(\bigcap_{\ell \in \mathbf{Pa}_i^a} \mathbf{Ch}_\ell^a\right) = \{i\},$$

*where $\mathbf{Pa}_i^z$ and $\mathbf{Ch}_i^z$ are the sets of parents and children of node $z_i$ in $G^z$, respectively, while $\mathbf{Ch}_\ell^a$ is the set of children of $a_\ell$ in $G^a$.*

We note that the above definition is slightly different from the original one, since the intersections run over $\mathbf{Ch}_i^z$, $\mathbf{Pa}_i^z$ and $\mathbf{Pa}_i^a$ instead of over some sets of indexes $\mathcal{I}, \mathcal{J} \subset \{1, ..., d_z\}$ and $\mathcal{L} \subset \{1, ..., d_a\}$. This slightly simplified criterion is equivalent to the original one, which we now demonstrate for the interested reader.

**Proposition 24.** *Let $G = [G^z\ G^a] \in \{0,1\}^{d_z \times (d_z + d_a)}$. The criterion of Def. 23 holds for $G$ if and only if the following holds for $G$: For all $i \in \{1, ..., d_z\}$, there exist sets $\mathcal{I}, \mathcal{J} \subset \{1, ..., d_z\}$ and $\mathcal{L} \subset \{1, ..., d_a\}$ such that*

$$\left(\bigcap_{j \in \mathcal{I}} \mathbf{Pa}_j^z\right) \cap \left(\bigcap_{j \in \mathcal{J}} \mathbf{Ch}_j^z\right) \cap \left(\bigcap_{\ell \in \mathcal{L}} \mathbf{Ch}_\ell^a\right) = \{i\},$$

*Proof.* The direction "$\Longrightarrow$" is trivial, since we can simply choose $\mathcal{I} := \mathbf{Ch}_i^z$, $\mathcal{J} := \mathbf{Pa}_i^z$ and $\mathcal{L} := \mathbf{Pa}_i^a$.

To show the other direction, we notice that we must have $\mathcal{I} \subset \mathbf{Ch}_i^z$, $\mathcal{J} \subset \mathbf{Pa}_i^z$ and $\mathcal{L} \subset \mathbf{Pa}_i^a$, otherwise one of the sets in the intersection would not contain $i$, contradicting the criterion. Thus, the criterion of Def. 23 intersects the same sets or more sets. Moreover these potential additional sets must contain $i$ because of the obvious facts that $j \in \mathbf{Ch}_i^z \iff i \in \mathbf{Pa}_j^z$ and $\ell \in \mathbf{Pa}_i^a \iff i \in \mathbf{Ch}_\ell^a$, thus they do not change the result of the intersection. $\square$

We can now derive the fact that, if all assumptions of Thm. 8 and the graphical criterion of Def. 23 hold, then the learned representation will be completely disentangled:

**Proposition 25** (Complete disentanglement as a special case). *Suppose all assumptions of Thm. 8 and the graphical criterion of Def. 23. Then, $\hat{\theta}$ is completely disentangled, i.e. $\hat{\theta}$ and $\theta$ are permutation-equivalent.*

*Proof.* Since assumptions of Thm. 8 holds, we have that $\theta$ and $\hat{\theta}$ are equivalent up to $CP^\top$ where $C$ is $G^z$-consistent, $(G^z)^\top$-consistent and $G^a$-consistent. Using Lemma 17, we have that, for all $i$,

$$(C_{i,\cdot})^\top \in \mathbb{R}^{d_z}_{\bigcap_{j \in G^z_{i,\cdot}} G^z_{\cdot,j}} \cap \mathbb{R}^{d_z}_{\bigcap_{j \in ((G^z)^\top)_{i,\cdot}} ((G^z)^\top)_{\cdot,j}} \cap \mathbb{R}^{d_z}_{\bigcap_{j \in G^a_{i,\cdot}} G^a_{\cdot,j}} \tag{117}$$

$$= \mathbb{R}^{d_z}_{\left(\bigcap_{j \in G^z_{i,\cdot}} G^z_{\cdot,j}\right) \cap \left(\bigcap_{j \in G^z_{\cdot,i}} G^z_{j,\cdot}\right) \cap \left(\bigcap_{j \in G^a_{i,\cdot}} G^a_{\cdot,j}\right)} \tag{118}$$

$$= \mathbb{R}^{d_z}_{\left(\bigcap_{j \in \mathbf{Pa}^z_i} \mathbf{Ch}^z_j\right) \cap \left(\bigcap_{j \in \mathbf{Ch}^z_i} \mathbf{Pa}^z_j\right) \cap \left(\bigcap_{\ell \in \mathbf{Pa}^a_i} \mathbf{Ch}^a_\ell\right)} \tag{119}$$

$$= \mathbb{R}^{d_z}_{\{i\}} . \tag{120}$$

Thus $C$ is in fact a diagonal matrix, and hence $\hat{\theta}$ is completely disentangled. $\qquad\square$

### A.2.8 Interpreting the meaning of Theorem 8 and the $\sim_{\text{con}}$-equivalence (Def. 7)

To interpret the conclusion of Thm. 8, which is that the learned model $\hat{\theta}$ is consistent to the ground-truth model $\theta$, i.e. $\theta \sim_{\text{con}} \hat{\theta}$ (Def. 7), we recall the example introduced in Sec. 3.2.1: Consider the case where the ground-truth graphs $G^z = \mathbf{0}$ (no temporal dependencies) and $G^a$ is

$$G^a = \begin{bmatrix} 1 & & & & \\ 1 & & & & \\ & 1 & & & \\ & 1 & & & \\ & & 1 & & \\ & & 1 & & \\ & & & 1 & \\ & & & 1 & \\ 1 & & & & 1 \\ 1 & & & & 1 \end{bmatrix},$$

which does not satisfy the graphical criterion of Def. 23. Then, $\theta \sim_{\text{con}} \hat{\theta}$ implies that: (i) $\hat{G}$ is the same as $G$, up to a permutation, and (ii) both representations $\mathbf{f}^{-1}$ and $\hat{\mathbf{f}}^{-1}$ are linked by a linear transformation $L = CP^\top$ (assuming $\mathbf{T}(z) := z$ for simplicity) where the matrix $C$ is $G^z$-consistent, $(G^z)^\top$-consistent and $G^a$-consistent. The conditions of $G^z$-consistency and $(G^z)^\top$-consistency are vacuous, since $[\mathbb{1} - \mathbf{0}(\mathbb{1} - \mathbf{0})^\top]^+ = \mathbb{1}$, i.e. they do not enforce anything on $C$. However, $G^a$-consistence forces $C$ to have the same zeros as

$$[\mathbb{1} - G^a(\mathbb{1} - G^a)^\top]^+ = \begin{bmatrix} 1 & 1 & & & & & & & 1 & 1 \\ 1 & 1 & & & & & & & 1 & 1 \\ & & 1 & 1 & & & & & & \\ & & 1 & 1 & & & & & & \\ & & & & 1 & 1 & & & & \\ & & & & 1 & 1 & & & & \\ & & & & & & 1 & 1 & & \\ & & & & & & 1 & 1 & & \\ & & & & & & & & 1 & 1 \\ & & & & & & & & 1 & 1 \end{bmatrix}. \tag{121}$$

Lemma 17 gives a different perspective by telling us that $C$ being $G^a$-consistent is equivalent to having $C_{i,j} = 0$ whenever $j \notin \bigcap_{\ell \in G^a_{i,\cdot}} G^a_{\cdot,\ell}$, which is equivalent to having $j \in \bigcup_{\ell \in \mathbf{Pa}^a_i} (\mathbf{Ch}^a_\ell)^c$. This allows us to see that *the ground-truth factor $z_i$ is not a function of the learned factor $\hat{z}_j$ ($C_{i,j} = 0$) whenever there exists an action $a_\ell$ that targets $z_i$, but not $z_j$.*

## B EXPERIMENTS

### B.1 SYNTHETIC DATASETS

We now provide a detailed description of the synthetic datasets used in experiments of Sec. 4, which exactly match those of Lachapelle et al. [2022], except for the graphs used. We nevertheless provide a full description of the datasets used here for completeness.

For all experiments, the dimensionality of $X^t$ is $d_x = 20$ and the ground-truth $\mathbf{f}$ is a random neural network with three hidden layers of 20 units with Leaky-ReLU activations with negative slope of 0.2. The weight matrices are sampled according to a 0-1 Gaussian distribution and, to make sure $\mathbf{f}$ is injective as assumed in all theorems of this paper, we orthogonalize its columns. Inspired by typical weight initialization in NN [Glorot and Bengio, 2010], we rescale the weight matrices by $\sqrt{\frac{2}{1+0.2^2}}\sqrt{\frac{2}{d_{in}+d_{out}}}$. The standard deviation of the Gaussian noise added to $\mathbf{f}(z^t)$ is set to $\sigma = 10^{-2}$ throughout. All datasets consist of 1 million examples.

We now present the different choices of ground-truth $p(z^t \mid z^{<t}, a^{<t})$ we explored in our experiments. In all cases considered, it is a Gaussian with covariance $0.0001I$ independent of $(z^{<t}, a^{<t})$ and a mean given by some function $\mu(z^{t-1}, a^{t-1})$ carefully chosen to satisfy the assumptions of Thm. 8. Notice that we hence are in the case where $k = 1$ which is not covered by the theory of Khemakhem et al. [2020a]. We suppose throughout that $d_z = 10$ and $d_a = 5$. In all *time-sparsity* experiments, sequences have length $T = 2$. In *action-sparsity* experiments, the value of $T$ has no consequence since we assume there is no time dependence.

**Transition function of the time-sparsity datasets (left of Table 1).** The mean function in this case is given by

$$
\mu(z^{t-1}, a^{t-1}) := z^{t-1} + 0.5 \begin{bmatrix} G_1^z \cdot \sin(\frac{3}{\pi} z^{t-1}) \\ G_2^z \cdot \sin(\frac{4}{\pi} z^{t-1} + 1) \\ \vdots \\ G_{d_z}^z \cdot \sin(\frac{d_z+2}{\pi} z^{t-1} + d_z - 1) \end{bmatrix}, \tag{122}
$$

where $G_i^z$ is the $i$th row of the ground-truth causal graph $G^z$, the $\sin$ function is applied element-wise, the $\cdot$ is the dot product between two vectors and the summation in the $\sin$ function is broadcasted. The various frequencies and phases in the $\sin$ functions ensures the sufficient time-variability assumption of Thm. 8 is satisfied.

**Graphs of the datasets with temporal dependence (left of Table 1).**

$$
G_{(1)}^z := \begin{bmatrix} 1 & 1 & & & & & & & & \\ 1 & 1 & & & & & & & & \\ & & 1 & 1 & & & & & & \\ & & 1 & 1 & & & & & & \\ & & & & 1 & 1 & & & & \\ & & & & 1 & 1 & & & & \\ & & & & & & 1 & 1 & & \\ & & & & & & 1 & 1 & & \\ & & & & & & & & 1 & 1 \\ & & & & & & & & 1 & 1 \end{bmatrix} \quad G_{(2)}^z := \begin{bmatrix} 1 & 1 & & & & & & & 1 & 1 \\ 1 & 1 & & & & & & & 1 & 1 \\ & & 1 & 1 & & & & & & \\ & & 1 & 1 & & & & & & \\ & & & & 1 & 1 & & & & \\ & & & & 1 & 1 & & & & \\ & & & & & & 1 & 1 & & \\ & & & & & & 1 & 1 & & \\ 1 & 1 & & & 1 & 1 & & & 1 & 1 \\ 1 & 1 & & & 1 & 1 & & & 1 & 1 \end{bmatrix} \tag{123}
$$

**Transition function of the action-sparsity datasets (right of Table 1).** The mean function is given by

$$
\mu(z^{t-1}, a^{t-1}) := \begin{bmatrix} G_1^a \cdot \sin(\frac{3}{\pi} a^{t-1}) \\ G_2^a \cdot \sin(\frac{4}{\pi} a^{t-1} + 1) \\ \vdots \\ G_{d_z}^a \cdot \sin(\frac{d_z+2}{\pi} a^{t-1} + d_z - 1) \end{bmatrix}, \tag{124}
$$

which is analogous to (122).

**Graphs of the datasets with actions (right of Table 1).**

$$
G^a_{(1)} := \begin{bmatrix} 1 & & & & \\ 1 & & & & \\ & 1 & & & \\ & 1 & & & \\ & & 1 & & \\ & & 1 & & \\ & & & 1 & \\ & & & 1 & \\ & & & & 1 \\ & & & & 1 \end{bmatrix} \quad G^a_{(2)} := \begin{bmatrix} 1 & & & \\ 1 & & & \\ & 1 & & \\ & 1 & & \\ & & 1 & \\ & & 1 & \\ & & & 1 \\ & & & 1 \\ 1 & & & 1 \\ 1 & & & 1 \end{bmatrix} \tag{125}
$$

## B.2 IMPLEMENTATION DETAILS OF THE CONSTRAINED VAE APPROACH

All details of our implementation matches those of Lachapelle et al. [2022] (except for the constrained optimization which is novel to our work). We nevertheless repeat all details here for completeness.

**Learned mechanisms.** Every coordinate $z_i$ of the latent vector has its own mechanism $\hat{p}(z_i^t \mid z^{<t}, a^{<t})$ that is Gaussian with mean outputted by $\hat{\mu}_i(z^{t-1}, a^{t-1})$ (a multilayer perceptron with 5 layers of 512 units) and a learned variance which does not depend on the previous time steps. For learning, we use the typical parameterization of the Gaussian distribution with $\mu$ and $\sigma^2$ and not its exponential family parameterization. Throughout, the dimensionality of $Z^t$ in the learned model always match the dimensionality of the ground-truth (same for baselines). Learning the dimensionality of $Z^t$ is left for future work.

**Prior of $Z^1$ in time-sparsity experiments.** In *time-sparsity* experiments, the prior of the first latent $\hat{p}(Z^1)$ (when $t = 1$) is modelled separately as a Gaussian with learned mean and learned diagonal covariance. Note that this learned covariance at time $t = 1$ is different from the subsequent learned conditional covariance at time $t > 1$.

**Learned graphs $\hat{G}^z$ and $\hat{G}^a$.** As explained in Sec. 3.3, to allow for gradient-based optimization, each edge $\hat{G}_{i,j}$ is viewed as a Bernoulli random variable with probability of success sigmoid($\gamma_{i,j}$), where $\gamma_{i,j}$ is a learned parameter. The gradient of the loss with respect to the parameter $\gamma_{i,j}$ is estimated using the Gumbel-Softmax Gradient estimator [Jang et al., 2017, Maddison et al., 2017]. We found that initializing the parameters $\gamma_{i,j}$ to a large value such that the probability of sampling all edge is almost one improved performance. In *time-sparsity* experiments, there is no action so $\hat{G}^a$ is fixed to 0, i.e. it is not learned. Analogously, in *action-sparsity* experiments, there is no temporal dependence so $\hat{G}^z$ is fixed to 0.

**Encoder/Decoder.** In all experiments, including baselines, both the encoder and the decoder is modelled by a neural network with 6 fully connected hidden layers of 512 units with LeakyReLU activation with negative slope 0.2. For all VAE-based methods, the encoder outputs the mean and a diagonal covariance. Moreover, $p(x|z)$ has a *learned* isotropic covariance $\sigma^2 I$. Note that $\sigma^2 I$ corresponds to the covariance of the independent noise $N^t$ in the equation $X^t = \mathbf{f}(Z^t) + N^t$.

**Constrained optimization.** Let ELBO($\hat{\mathbf{f}}, \hat{\boldsymbol{\lambda}}, \hat{G}, q$) be the ELBO objective evaluated on the whole dataset. The constrained optimization we want to solve is

$$
\max_{\hat{\mathbf{f}}, \hat{\boldsymbol{\lambda}}, \gamma, q} \mathbb{E}_{\hat{G} \sim \sigma(\gamma)} \text{ELBO}(\hat{\mathbf{f}}, \hat{\boldsymbol{\lambda}}, \hat{G}, q) \text{ subject to } \mathbb{E}_{\hat{G} \sim \sigma(\gamma)} ||\hat{G}||_0 \leq \beta . \tag{126}
$$

where $\hat{G} \sim \sigma(\gamma)$ means that $\hat{G}_{i,j}$ are independent and distributed according to $\sigma(\gamma_{i,j})$. Because $\mathbb{E}_{\hat{G} \sim \sigma(\gamma)} ||\hat{G}||_0 = ||\sigma(\gamma)||_1$ where $\sigma(\gamma)$ is matrix, the constraint becomes $||\sigma(\gamma)||_1 \leq \beta$. To solve this problem, we perform gradient descent-ascent on the Lagrangian function given by

$$
\mathbb{E}_{\hat{G} \sim \sigma(\gamma)} \text{ELBO}(\hat{\mathbf{f}}, \hat{\boldsymbol{\lambda}}, \hat{G}, q) - \alpha(||\sigma(\gamma)||_1 - \beta) \tag{127}
$$

where the ascent step is performed w.r.t. $\hat{\mathbf{f}}, \hat{\boldsymbol{\lambda}}, \hat{G}$ and $q$; and the descent step is performed w.r.t. Lagrangian multiplier $\alpha$, which is forced to remain greater or equal to zero via a simple projection step. As suggested by Gallego-Posada et al. [2021], we perform *dual restarts* which simply means that, as soon as the constraint is satisfied, the Lagrangian multiplier is reset to 0. We used the library `Cooper` [Gallego-Posada and Ramirez, 2022], which implement many constrained optimization procedure in Python, including the one described above. Note that we use Adam [Kingma and Ba, 2015] for the ascent steps and standard gradient descent for the descent step on the Lagrangian multiplier $\alpha$.

We also found empircally that the following schedule for $\beta$ is helpful: We start training with $\beta = \max_{\hat{G}} ||G||_0$ and linearly decreasing its value until the desired number of edges is reached. This avoid getting a sparse graph too quickly while training, thus letting enough time to the model parameters to learn. In each experiment, we trained for 300K iterations, and the $\beta$ takes 150K to reach to go from its initial value to its desired value.

## B.3 DETAILS ABOUT THE $R_{\text{con}}$ METRIC AND ITS RELATION TO MCC AND $R$

To evaluate whether the learned representation is consistent to the ground-truth (Def. 7), as predicted by Thm. 8, we came up with a novel metric, denoted by $R_{\text{con}}$. Computing $R_{\text{con}}$ goes as follows: First, we permute the learned representations $\hat{z}$ using the permutation $\hat{P}$ found by MCC (Sec. 4), i.e. $\hat{z}_{\text{perm}} := \hat{P}^{\top} \hat{z}$. Then, we compute the sparsity pattern imposed by the consistency equivalence (Def. 7), denoted by $S_C$. Then, for every $i$, we predict the ground-truth $z_i$ given only the factors allowed, i.e. $(S_C)_{i,.} \odot \hat{z}_{\text{perm}}$, and compute the associated coefficient of multiple correlations $R_{\text{con},i}$ and report the mean, i.e. $R_{\text{con}} := \frac{1}{d_z} \sum_{i=1}^{d_z} R_{\text{con},i}$. It is easy to see that we must have $R_{\text{con}} \leq R$, since $R_{\text{con}}$ was computed with less features than $R$. Moreover, MCC $\leq R_{\text{con}}$, because MCC can be thought of as computing exactly the same thing as for $R_{\text{con}}$, but by predicting $z_i$ only from $\hat{z}_{\text{perm},i}$, i.e. with less features than $R_{\text{con}}$.

This means we always have $0 \leq$ MCC $\leq R_{\text{con}} \leq R \leq 1$. This is a nice property which allows to compare all three metrics together.