# OpenReview forum: "Partial Disentanglement via Mechanism Sparsity"
_auai.org/UAI/2022/Workshop/CRL — CRL@UAI 2022 Oral_

### Official Review · Reviewer_n2P5 · 2022-06-27
**Useful theory extension, needs a related work section**

**Rating:** 6
**Confidence:** 3

**Review:**

**Paper summary:**
This paper extends the “disentanglement via mechanism sparsity” theory of Lapachelle et al. (2022) to allow for weaker notions of identifiability that fall between linear and permutation equivalence. The authors call this “partial disentanglement”.

**Review summary:**
The paper is well written and presented. Overall, I think it is worth accepting the paper as its theory extension/generalization could be useful. However, without a thorough related work section, I find the terms “complete” and “partial” disentanglement to cause much confusion given the number of existing notions of "partial disentanglement". Addressing this would significantly improve the paper.

**Pros**
- The writing is clear and easy to follow.
- The generalization of the theory of Lapachelle et al. (2022) could be useful to the community, and a good topic for discussion at the workshop.

**Cons**
- The introduced terms of “complete” and “partial” disentanglement cause confusion by overloading existing ideas/terms. Solutions could include:
    1. Using existing terms rather than introducing new ones, e.g. “We extend Lapachelle et al. (2022) to include not only permutation identifiability, but also linear identifiability and everything in between”. That is, sticking with the existing ideas of strong/permutation identifiability and weak/linear identifiability rather than introducing the more ambiguous terms of "complete" and "partial" disentanglement.
    2. Thoroughly discussing the introduced terms and how they relate to existing ones (see next point).
- Lacks a discussion of how “partial disentanglement” relates to existing such notions, e.g.:
    - Disentanglement metrics. For example, the D and C scores of Eastwood & Williams (2018) are 1 given "complete" disentanglement and in [0,1) given "partial" disentanglement.
    - Block disentanglement of Von Kuegelgen et al (2021). Perhaps there are many other forms of “partial” disentanglement, warranting a different name for the notion introduced in this paper?
    - Shu et al. (2020) decompose disentanglement into two distinct concepts: “consistency” and “restrictiveness” – does this “consistency” relate to the author’s “consistency”?
- The employed constraint is not really a contribution, as claimed.
    - Already used for achieving sparsity, e.g. Gallego-Posada et al. (2021).
    - No empirical evidence, just a somewhat weak point on easier hyperparameter searching due to increased interpretability.
    - Solution: tone down emphasis on the constraint being a *contribution* of the paper.


Eastwood, C., & Williams, C. K. I. (2018). A framework for the quantitative evaluation of disentangled representations. In *International Conference on Learning Representations*.

Shu, R., Chen, Y., Kumar, A., Ermon, S., & Poole, B. (2020). Weakly Supervised Disentanglement with Guarantees. In *International Conference on Learning Representations*.

---

### Official Review · Reviewer_jds9 · 2022-06-29

**Rating:** 8
**Confidence:** 3

**Review:**

Summary: This paper extends the theory of the disentalgement via mechanism sparsity introduced by [Lachapelle et al 2022] by relaxing the assumptions on the underline ground-truth graph. This paper develops a theory that guarantees partial disentalgement of the latent variables. In certain sense, the theory developed in this paper interpolates between [Khlemakhem et al 2020] and [Lachapelle et al 2022] in the following sense. The first paper provides sufficient conditions that allow learning $T(f^{-1}(x))$ up to a linear transformation and the second one proposes sufficient conditions to ensure that $T(f^{-1}(x))$ are identified up to a permutation and scaling. This paper introduces a new equivalence relation, called consistency, that is stronger than [Khlemakhem et al 2020] and weaker than the one in [Lachapelle et al 2022] and shows that under certain assumptions the model ground truth model can be recovered up to the consistency equivalence relation. This theory strictly generalizes [Lachapelle et al 2022] in the sense that the criterion proposed in that paper is a special case of the criterion proposed in this paper.

This paper also establishes some nice properties of the consistency relation. In particular, the set of matrices consistent with a given one forms a group under matrix multiplication operation.

The paper is well-written and appears to be correct (however I have not checked the proofs in the appendices carefully).

I think the biggest limitation of this work is that it requires the sufficient statistics of the distribution in the latent space to be 1-dimensional. In particular, that does not allow one to choose a general Gausssian Mixture or a general mixture of Gamma distributions in the latent space (a common choice in the literature). I think it will also be nice to include a more detailed discussion of Sufficient time-variability and Sufficient action-variability assumptions. Do they ``typically" hold?

I believe that this paper makes a significant contribution to the theory of causal representation learning.

---

### Meta-Review · Program_Chairs · 2022-07-06

**Recommendation:** Accept (Oral)
**Confidence:** 4

**Metareview:**

The reviewers agree that the paper makes interesting (mostly theoretical) contributions and is a good fit for the workshop.  They also provide several suggestions for further improving the work (multi-variate sufficient statistics, more extensive comparison with existing works and identifiability notions). I concur and recommend acceptance.

---

### Decision · Program_Chairs · 2022-07-06

Accept (Oral)